# NPC1-dependent alterations in K$_V$2.1−Ca$_V$1.2 nanodomains drive neuronal death in models of Niemann-Pick Type C disease

Maria Casas [1], Karl D. Murray[1,2], Keiko Hino[3], Nicholas C. Vierra[1], Sergi Simó [3], James S. Trimmer [1], Rose E. Dixon [1] & Eamonn J. Dickson [1] ✉

Lysosomes communicate through cholesterol transfer at endoplasmic reticulum (ER) contact sites. At these sites, the Niemann Pick C1 cholesterol transporter (NPC1) facilitates the removal of cholesterol from lysosomes, which is then transferred to the ER for distribution to other cell membranes. Mutations in NPC1 result in cholesterol buildup within lysosomes, leading to Niemann-Pick Type C (NPC) disease, a progressive and fatal neurodegenerative disorder. The molecular mechanisms connecting NPC1 loss to NPC-associated neuropathology remain unknown. Here we show both in vitro and in an animal model of NPC disease that the loss of NPC1 function alters the distribution and activity of voltage-gated calcium channels (Ca$_V$). Underlying alterations in calcium channel localization and function are K$_V$2.1 channels whose interactions drive calcium channel clustering to enhance calcium entry and fuel neurotoxic elevations in mitochondrial calcium. Targeted disruption of K$_V$2−Ca$_V$ interactions rescues aberrant Ca$_V$1.2 clustering, elevated mitochondrial calcium, and neurotoxicity in vitro. Our findings provide evidence that NPC is a nanostructural ion channel clustering disease, characterized by altered distribution and activity of ion channels at membrane contacts, which contribute to neurodegeneration.

Cholesterol, an essential sterol, is particularly abundant in the brain[1,2] where it fulfils important biophysical and signaling functions that affect ion permeability, cellular signaling, and general programs of transcription (for general review see refs. 1,3–7). Given the importance of maintaining cholesterol homeostasis for controlled neuronal function, it is not surprising that mutations that disrupt cholesterol production or transport result in severe abnormalities of the central nervous system (CNS) and neurodegeneration[8,9].

In the brain, all cholesterol is made locally[10] with neurons drawing their homeostatic requirements from two sources (i) de novo synthesis in the Endoplasmic Reticulum (ER), and (ii) uptake of the externally derived cholesterol-conjugated lipoproteins. External cholesterol is imported into neurons in a receptor-dependent manner and trafficked to late endosomes /lysosomes[11]. At these vesicular structures, the Niemann Pick type C (NPC) proteins are key players in mobilizing cholesterol to the ER and other cellular membranes[12,13]. Specifically, following liberation by acidic lipases in the lysosome lumen, cholesterol binds NPC2, which subsequently interacts with the transmembrane NPC1 cholesterol transporter to facilitate the egress of cholesterol across the lysosome membrane to its cytoplasmic leaflet before subsequent transfer to the ER at lysosome−ER Membrane Contact Sites (MCSs). The importance of the NPC1 protein is clear; physiologically, it is a key gatekeeper for cholesterol homeostasis and in doing so tunes mTORC1 activity, lipid transfer at membrane contact sites, and Ca$^{2+}$ signaling[14–17]. Pathophysiologically, missense mutations in NPC1 lead to the progressive neurodegenerative disorder NPC

[1]Department of Physiology and Membrane Biology, School of Medicine, University of California, Davis, CA, USA. [2]Department of Psychiatry & Behavioral Sciences, School of Medicine, University of California, Davis, CA, USA. [3]Department of Cell Biology and Human Anatomy, School of Medicine, University of California, Davis, CA, USA. ✉e-mail: ejdickson@ucdavis.edu

disease. This terminal condition has no cure and is characterized by the progressive neurodegeneration of several brain regions and a host of devastating symptoms including seizures, psychiatric problems, and dementia. Despite its ubiquitous expression, key role in cholesterol transport, and fundamental requirement for human health, the molecular mechanism(s) linking loss of NPC1 function to neurodegeneration are unknown.

A key mechanism through which the lysosome communicates instructions to other organelles is through physical membrane contacts. These intracellular synapses are intimate sites (~10–30 nm) of membrane contact between two or more organelles[18–22] that use lipids and $Ca^{2+}$ as "currency" to communicate information. A key signaling lipid that is transferred at ER–lysosome MCSs is cholesterol. The NPC1 protein not only influences cholesterol transport at lysosome–ER MCSs but is also a master regulator of other MCSs within mammalian cells[14,16,17,23,24]. Loss of function or disease mutations in NPC1 cholesterol transporter results in the remodeling of ER–lysosome[14,15], ER–Golgi[16] and ER–Mitochondria[17] MCSs to influence mTORC1 signaling, anterograde trafficking to the Plasma Membrane (PM), and mitochondrial $Ca^{2+}$, respectively. That said, there is little information as to how lysosomal cholesterol transport alters the molecular choreography of neuronal ER–PM junctions. This is important because (i) 40–90% of cellular cholesterol is found in the PM[7,25], therefore disruption of NPC1-mediated cholesterol homeostasis would be expected to result in modification of ER–PM MCSs, and (ii) ER–PM MCSs are critical platforms for regulating lipid and $Ca^{2+}$ homeostasis in neurons[2,26–31]. Despite such importance, we do not know how the loss of cholesterol transport from the lysosome alters ER–PM MCSs, which proteins within these contacts have altered distribution/activity, and the cellular consequences of such changes.

In neurons, a key regulator of ER–PM MCSs is the voltage-gated potassium channel $K_V2.1$. Gating of this voltage-dependent ion channel provides an inhibitory brake on excitability[32,33]. However, over the past few years it has been appreciated that a subset of non-conducting $K_V2$ ion channels perform a physical function by generating ER–PM junctions through an association between their C-terminus and ER-localized VAP proteins in a phosphorylation-dependent manner[26,34,35]. Moreover, at ER–PM MCSs, $K_V2$ channels physically interact with and cluster $Ca^{2+}$ signaling proteins like voltage-gated L-type $Ca^{2+}$ channels ($Ca_V1$)[2,31,36]. $Ca_V1$ channels are key neuronal $Ca^{2+}$ signaling proteins which serve as a principal route of $Ca^{2+}$ entry at depolarized membrane potentials. Such biophysical properties give $Ca_V1$ a prominent role in neuronal excitability and gene expression, which are crucial for neuronal signaling and synaptic plasticity[37]. Moreover, several neurodegenerative diseases are characterized by excessive $Ca^{2+}$ influx through $Ca_V1$ channels[38,39] with their targeted knockdown or inhibition being neuroprotective[40–42]. While $K_V2$-mediated targeting and regulation of $Ca_V1$ channels at ER–PM MCSs serves a physiologically important role in neurons, there is no information on how this form of regulation may be altered in disease, including how lysosomal cholesterol may regulate $K_V2$ clustering to tune excitability and $Ca^{2+}$ signaling at ER–PM MCSs to impact neurodegeneration in NPC disease.

In this current study, we report that loss of NPC1 function results in phosphorylation-dependent increases in $K_V2.1$ clustering at the PM. We show that enhanced Kv2.1-containing ER-PM MCSs are enriched in SERCA, $Ca_V1.2$, and RyR resulting in enhanced $Ca^{2+}$ signaling at $K_V2.1$ associated ER-PM nanodomains. In addition to remodeling of ER–PM MCSs, we also show that $Ca^{2+}$ handling proteins at ER–Mitochondrial MCSs are modified following loss of NPC1 function. Collaboratively, this molecular transformation of MCSs leads to deviant elevations in mitochondrial $Ca^{2+}$ and neurotoxicity. Importantly, we demonstrate that uncoupling upstream $K_V2$–$Ca_V1$ interactions rescues mitochondrial $Ca^{2+}$ and neurotoxicity and thus represents a potential therapeutic target in NPC1 disease. Collectively, these data demonstrate

that NPC is a nanostructural ion channel clustering disease with altered ion channel distribution/activity at MCSs contributing to neurodegeneration.

## Results

### Loss of NPC1 function increases voltage-dependent $Ca^{2+}$ entry and $Ca_V1$ channel clustering in isolated neurons

The opening of voltage-gated $Ca^{2+}$ channels ($Ca_V$) permits movement of $Ca^{2+}$ down its concentration gradient to increase intracellular $Ca^{2+}$ levels and represents a major mechanism through which neurons translate electrical signals into biochemical ones. Elevations in $Ca^{2+}$ are tightly regulated in mammalian neurons[43,44] with dysregulation linked to several neuropathologies[45–51]. Recently, we have reported that NPC1 inhibition, NPC1 mutations ($NPC1^{I1061T}$) or NPC1 knockout ($NPC1^{−/−}$) neurons are hyperexcitable, firing significantly more Action Potentials (APs) compared to age and sex-matched Wild-Type (WT) littermates[52]. Given the importance of the membrane potential for controlling $Ca^{2+}$ entry through $Ca_V$ channels[53], we wanted to test if NPC1 loss of function alters voltage-dependent $Ca^{2+}$ activity. To test this hypothesis, we incubated cortical neurons overnight using a specific pharmacological inhibitor of NPC1[54,55] (U18666A; referred to as U18 hereafter), which increases spontaneous and stimulated voltage responses as well as internal cholesterol accumulation (Figure S1A)[52] and loaded cells with the cytosolic $Ca^{2+}$ dye, Fluo-4. Quantitative analysis from Fluo-4 time-series experiments revealed that U18 treatment increased the amplitude and frequency of both spontaneous and stimulated (40 V, 1 Hz) elevations in Fluo-4 intensity relative to controls (Fig. 1A–D). Interestingly, the relative change in Fluo-4 intensity between basal activity and following stimulation was decreased compared to control ($\Delta F_{40V\ 1Hz}/\Delta F_{0-25s}$ in Fig. 1D), suggesting a narrower dynamic range for $Ca^{2+}$ signaling in NPC. Furthermore, preventing AP generation by treatment with the voltage-gated sodium channel blocker Tetrodotoxin (TTX) not only abrogated any NPC1-driven increase in basal and electrically evoked Fluo-4 intensities, but also reduced both spontaneous and stimulated cytosolic $Ca^{2+}$ amplitude elevations compared to control neurons (Fig. S1B).

Increased frequency of voltage-dependent $Ca^{2+}$ entry is likely driven by the hyperexcitability phenotype in NPC disease[52]. However, increases in the amplitude of $Ca^{2+}$ events suggested to us that in addition to enhanced hyperexcitability, $Ca_V$ channel expression and/or distribution may be altered following the loss of NPC1 function. A salient feature of $Ca_V1$ channels in neurons, cardiac myocytes, and smooth muscle is that their activity is regulated by clustering (for review see ref. 56), such that adjacent channels can functionally interact with one another permitting cooperative gating[57–61]. As a result, the activity of physically interacting channels in a cluster is driven by the highest open probability channel, thus enhanced channel clustering facilitates the amplification of $Ca^{2+}$ currents. To test if NPC1 loss of function alters $Ca_V$ distribution we treated cortical with U18 before fixing and immunolabeling for $Ca_V1.2$ and performing super-resolution AiryScan or single molecule localization TIRF microscopy (super-res$_{TIRF}$). Quantification of total intensity, cluster density, and cluster size from AiryScan confocal regions close to the PM demonstrated that U18-treatment enhanced $Ca_V1.2$ clustering in both the soma and dendrites (Fig. 2A). Analysis of super-res$_{TIRF}$ localization maps (resolution of 30 nm[62]) further corroborated our AiryScan analysis that U18-treatment increased the size and number of PM $Ca_V1.2$ clusters (Fig. 2B). Moreover, nearest neighbor cluster distance analysis revealed that $Ca_V1.2$ clusters are in closer spatial proximity following NPC1 loss of function (Fig. 2B).

In addition to $Ca_V1.2$, cortical neurons have other L- and P/Q-type $Ca_V$ channels that contribute to voltage-dependent $Ca^{2+}$ entry. To test if these $Ca_V$ channel isoforms are also altered, we immunolabeled cortical neurons with antibodies against $Ca_V1.3$ or $Ca_V2.1$ and

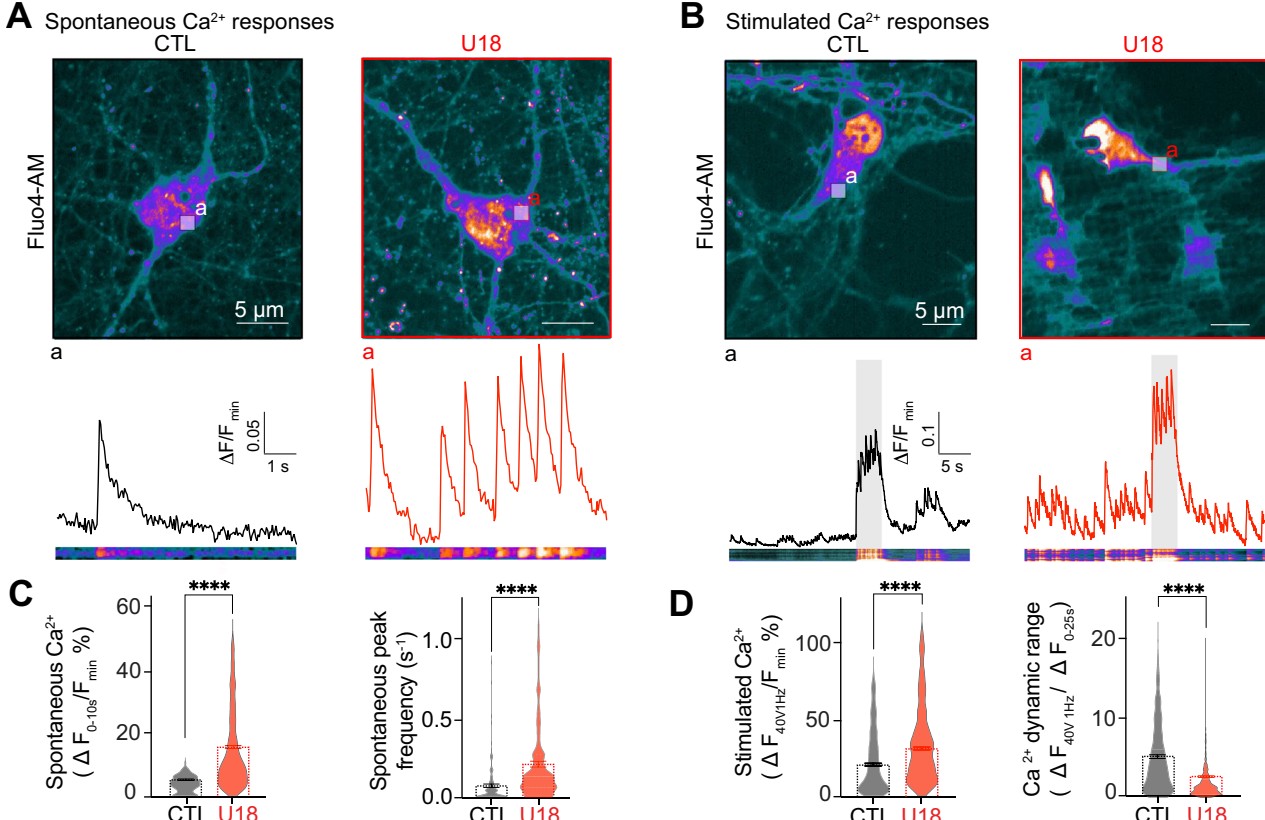

**Fig. 1 | Loss of NPC1 function increases voltage-dependent Ca²⁺ entry. A** *Top*, representative images of live CTL (black) and U18-treated (red) neurons loaded with fluo-4. *Bottom*, intensity time series and kymographs of spontaneous activity taken from the white region of interest (*a*) in both images. **B** *Top*, representative images of live CTL (black) and U18-treated (red) neurons loaded with fluo-4. *Bottom*, intensity time series and kymographs of stimulated activity taken from the white region of interest (*a*) in both images. Electrical stimulation (gray bar) was performed from seconds 26 to 31 at 40 V 1 Hz. **C** Quantification of spontaneous Ca²⁺ peak amplitude and frequency in CTL (black) and U18-treated (red) neurons.

$n = 108$ (CTL) and $n = 148$ (U18) neurons, and $n = 138$ (CTL) and $n = 683$ (U18) peaks were analyzed over 2 independent isolations. **D** Quantification of Ca²⁺ peak amplitude and Ca²⁺ dynamic range (relative change in Fluo-4 intensity between basal activity and following stimulation) in CTL (black) vs U18-treated (red) neurons. $N = 457–406$ (CTL) and $n = 656–707$ (U18) Ca²⁺ peaks, respectively, over 2 independent isolations. All error bars represent SEM. Statistical significance was calculated using Mann–Whitney $t$ test (two-tailed) in (**C**) and (**D**). ****$P < 0.0001$. CTL is control and U18 is U18666A.

quantified their distributions at the PM using super-resolution microscopy. Ca_V1.3 cluster sizes were significantly increased upon U18 treatment while the total fluorescence intensity, number of clusters and spatial proximity between adjacent channels was unaltered in the somatic region of neurons (Fig. 2C, D). Unlike the soma, Ca_V1.3 cluster size and density in dendrite regions remained unchanged while total fluorescence intensity significantly decreased (Fig. 2C). In contrast to Ca_V1.2 and Ca_V1.3, quantification of Ca_V2.1 revealed a significant reduction in cluster size and intensity with no significant difference in cluster number at the soma, while Cav2.1 clustering in dendrites remained unaltered by U18 (Fig. S2). Collectively, these data provide evidence that loss of NPC1 function facilitates clear and broad alterations in Ca_V1 channel distribution with significant increases in the size of Ca_V1.2 and Ca_V1.3 channel clusters in the cell soma.

### K_V2.1 clustering is enhanced following loss of NPC function in vitro and in animal models of NPC disease

Having demonstrated that the NPC1 cholesterol transporter influences the distribution of PM Ca_V channels, we next wanted to determine the molecular mechanism that increases somatic Ca_V1.2 and Ca_V1.3 channel clustering. To begin, we tested if a simple increase in Ca_V1.2 protein levels could account for increases in channel clustering. To that end, we extracted protein from cortical neurons cultured for 6-8 Days In Vitro (DIV) and found that total Ca_V1.2 protein levels were similar between control and U18-treated neurons (Fig. S3A) suggesting

a simple increase in protein expression is unlikely to underlie increases in PM Ca_V channel distribution.

Recently, voltage-gated K_V2.1 potassium channels have been demonstrated to organize neuronal Ca_V1 channels, impacting their distribution and activity in specific somatic microdomains[2,31]. Based on our observations that somatic Ca_V1.2 and Ca_V1.3 channel clusters are increased in NPC disease conditions, we tested if K_V2.1 channels also had altered distribution. Confocal and super-res_TIRF analysis from fixed cortical neurons immunolabeled for K_V2.1 revealed that the total fluorescence intensity, number of clusters, and cluster size were all increased in the PM from somatic and dendritic regions of U18-treated neurons relative to control (Fig. 3A). Additionally, in-depth analysis of super-res_TIRF maps determined that not only are K_V2.1 channel clusters larger, but they are closer together with U18 decreasing nearest neighbor distance between adjacent K_V2.1 channel clusters (Fig. 3B). Like Ca_V1.2 channels, total levels of K_V2.1 expression remained constant between control and U18 conditions (Fig. S3A).

Next, to test if animal models of NPC1 disease also have altered neuronal K_V2.1 distribution, we fixed, sectioned, and immunolabeled brain sections from WT and NPC1^II061T (most prevalent disease mutation) mice for K_V2.1. Similar to treatment with U18, mice harboring the NPC1^II061T mutation displayed increased K_V2.1 clustering in cortical (Fig. 3C) and hippocampal (Fig. S3B) pyramidal neurons, and cerebellar Purkinje neurons (Fig. 4D). Therefore, like Ca_V1.2, K_V2.1 clusters are larger and more abundant in NPC disease.

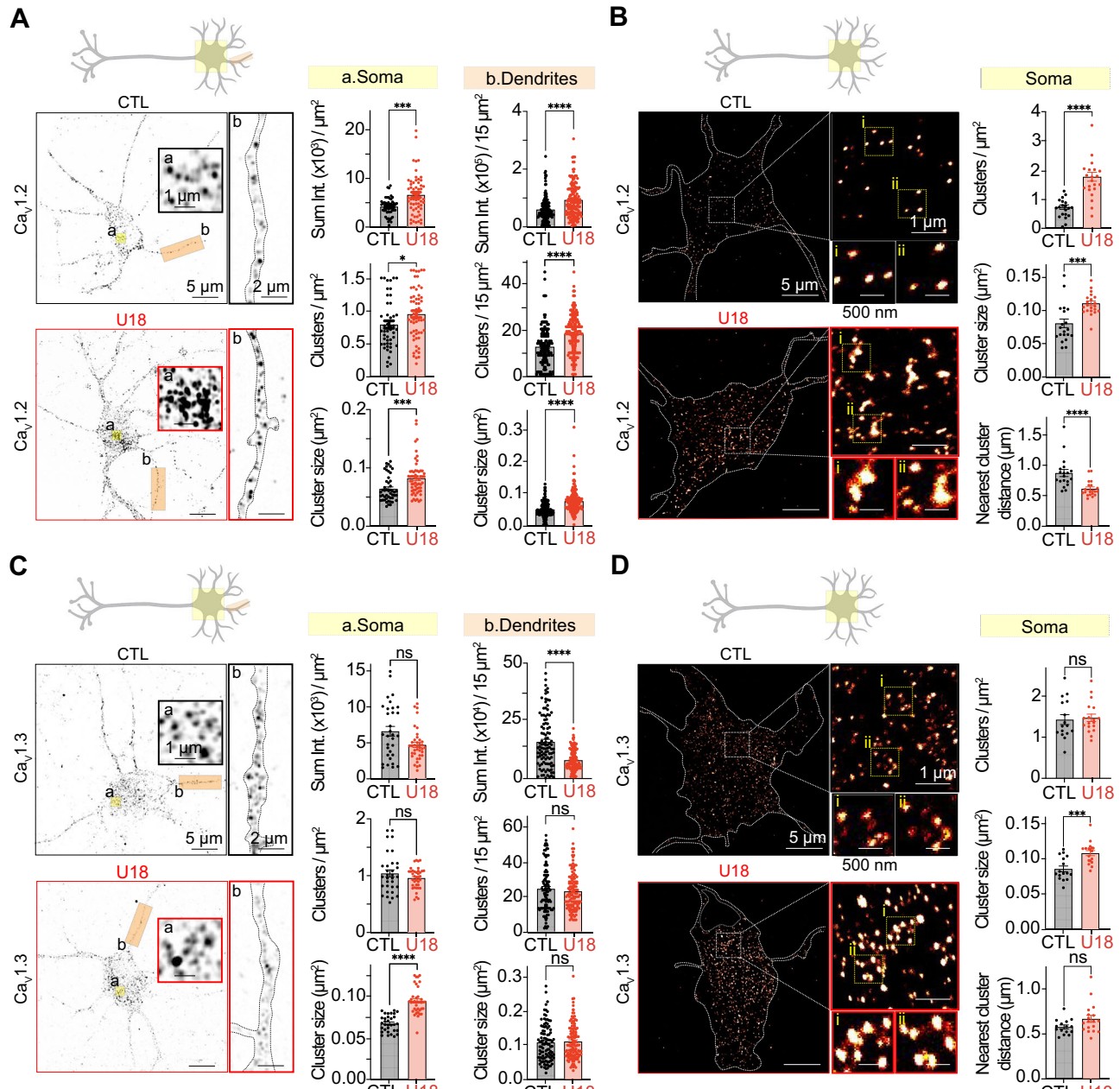

**Fig. 2 | Loss of NPC1 function alters voltage-dependent Ca$_V$1 channel distribution. A** Left: Representative super-resolution Airyscan images taken at a focal plane near the plasma membrane (PM) from CTL (black) and U18-treated (red) neurons fixed and immunolabeled for Ca$_V$1.2. *Right*, quantification of PM Ca$_V$1.2 total intensity, cluster density and cluster size of CTL (black) and U18-treated (red) neurons in the soma (left, yellow) and dendrite (right, orange) regions. $N = 46–48$ (CTL) and $n = 58–59$ (U18) neurons, and $n = 145$ (CTL) and $n = 161$ (U18) dendrites were analyzed across 5 independent isolations. **B** *Left*, representative super-resolution TIRF localization maps of CTL (black) and U18-treated (red) neurons fixed and immunolabeled for Ca$_V$1.2. *Right*, quantification of PM Ca$_V$1.2 cluster

density, cluster size, and nearest cluster distance of CTL (black) and U18-treated (red) neurons in the soma region. $N = 20$ (CTL) and $n = 21$ (U18) neurons were analyzed across 3 independent isolations. **C** Same as (**A**) only neurons fixed and immunolabeled for Ca$_V$1.3. $N = 32$ (CTL) and $n = 38$ (U18) neurons, and $n = 104$ (CTL) and $n = 123$ (U18) dendrites were analyzed across 3 independent isolations. **D** Same as B, only immunolabeled for Ca$_V$1.3. $N = 16$ (CTL) and $n = 18$ (U18) neurons were analyzed across 2 independent isolations. All error bars represent SEM. Statistical significance was calculated using Mann–Whitney (two-tailed) and Unpaired *t* tests (two-tailed) in (**A**)–(**D**). ns: not significant; *$P < 0.05$; **$P < 0.01$; ***$P < 0.001$; ****$P < 0.0001$. CTL is control and U18 is U18666A.

## NPC1 loss of function increases K$_V$2.1–Ca$_V$1.2 colocalization in vitro and in vivo

Loss of NPC1 function increases Ca$_V$1.2 (Fig. 2) and K$_V$2.1 (Fig. 3) clustering in somatic regions of neurons. Given that these two channels physically interact[31], we used two complementary approaches to test if loss of NPC1 function alters the proximity of K$_V$2.1–Ca$_V$1.2 channel complexes. First, using super-res$_{TIRF}$ imaging we quantified the relative distribution between Ca$_V$1.2 and K$_V$2.1 channel complexes. Figure 4A

shows that around 0.05% of neuronal somatic pixels were positive for both Ca$_V$1.2 and K$_V$2.1 in control conditions at this enhanced resolution. Inhibition of NPC1 led to a 3-fold increase in the number of somatic pixels positive for both Ca$_V$1.2 and K$_V$2.1, with increases in the average size of Ca$_V$1.2–K$_V$2.1 clusters (Fig. 4A). Furthermore, NPC1 loss-of-function also reduced the average distance between adjacent Ca$_V$1.2–K$_V$2.1 hetero-clusters (Fig. 4A). Identical results were observed in blinded experimental datasets (Fig. S4A, B) and NPC1$^{-/-}$ cells

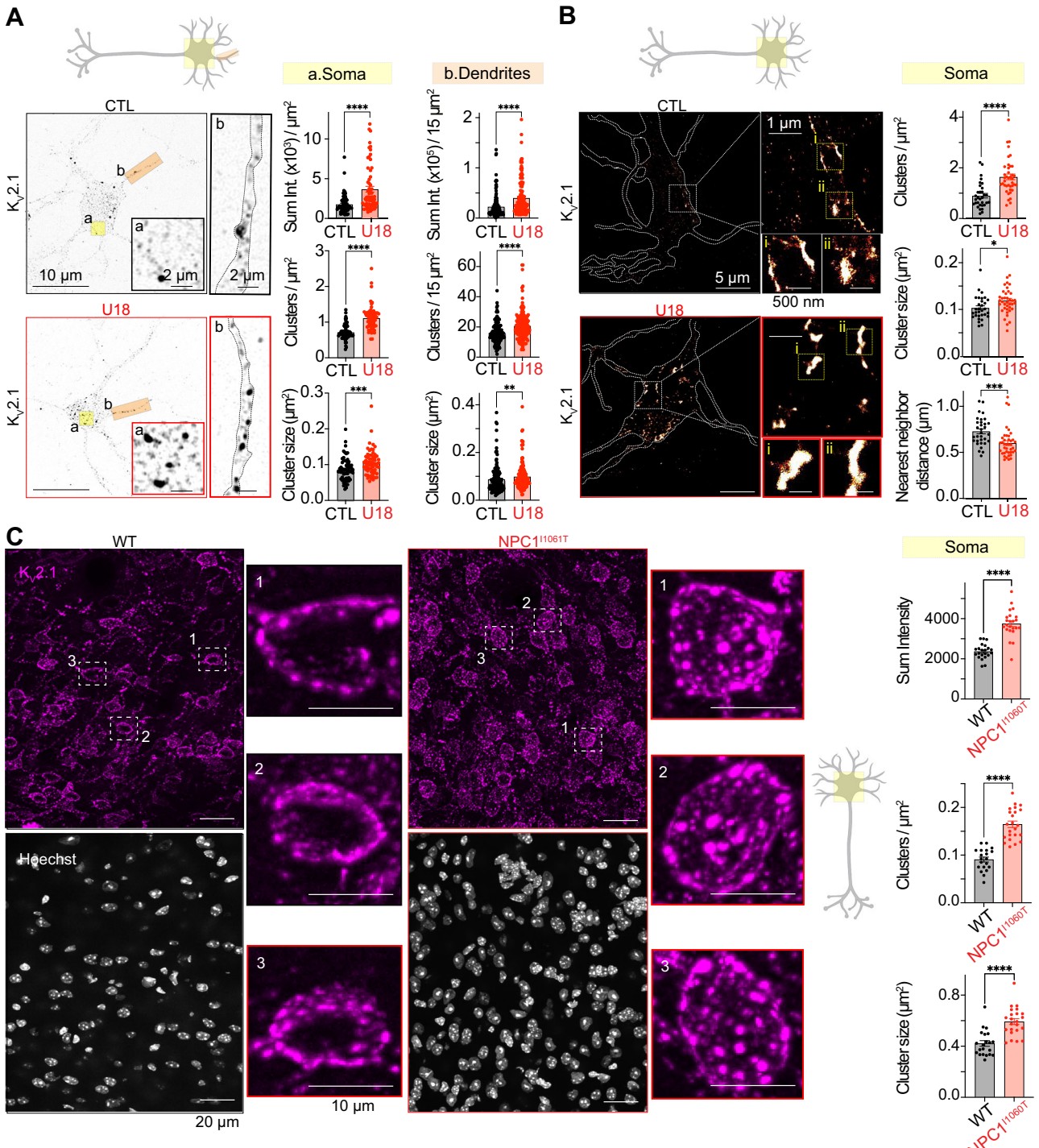

**Fig. 3 | Neurons lacking functional NPC1 have altered distribution of plasma membrane K$_V$2.1. A** *Left*, representative super-resolution Airyscan images taken at a focal plane near the PM of CTL (black) and U18-treated (red) neurons fixed and immunolabeled for K$_V$2.1. *Right*, quantification of PM K$_V$2.1 total intensity, cluster density and cluster size of CTL (black) and U18-treated (red) neurons in the soma (left, yellow) and dendrite (right, orange) regions. $N = 61$ (CTL) and $n = 64$ (U18) neurons and $n = 143$ (CTL) and $n = 162$ (U18) dendrites were analyzed across 5 independent isolations. **B** *Left*, representative super-resolution TIRF images of CTL (black) and U18-treated (red) neurons fixed and immunolabeled for K$_V$2.1. *Right*, quantification of PM K$_V$2.1 cluster density, cluster size and nearest cluster distance of CTL (black) and U18-treated (red) neurons in the soma region. N = 32 (CTL) and n = 40 (U18) neurons were analyzed across 5 independent isolations. **C** *Left*, representative maximum intensity projections from Wild-Type (WT) (black) and NPC1$^{I1061T}$ (red) cortical sagittal sections stained with Hoechst and immunolabeled for K$_V$2.1. *Right*, quantification of K$_V$2.1 total intensity, cluster size and cluster density of WT (black) and NPC1$^{I1061T}$ (red) neurons in the soma region. N = 20 – 22 (WT) and n = 23 (U18) neurons were analyzed across 2 pairs of animals. All error bars represent SEM. Statistical significance was calculated using Mann-Whitney (two-tail) and unpaired *t* test (two-tail) in (**A**)−(**C**); ns: not significant; *$P < 0.05$; **$P < 0.01$; ***$P < 0.001$; ****$P < 0.0001$. CTL is control, WT is wild-type, and U18 is U18666A.

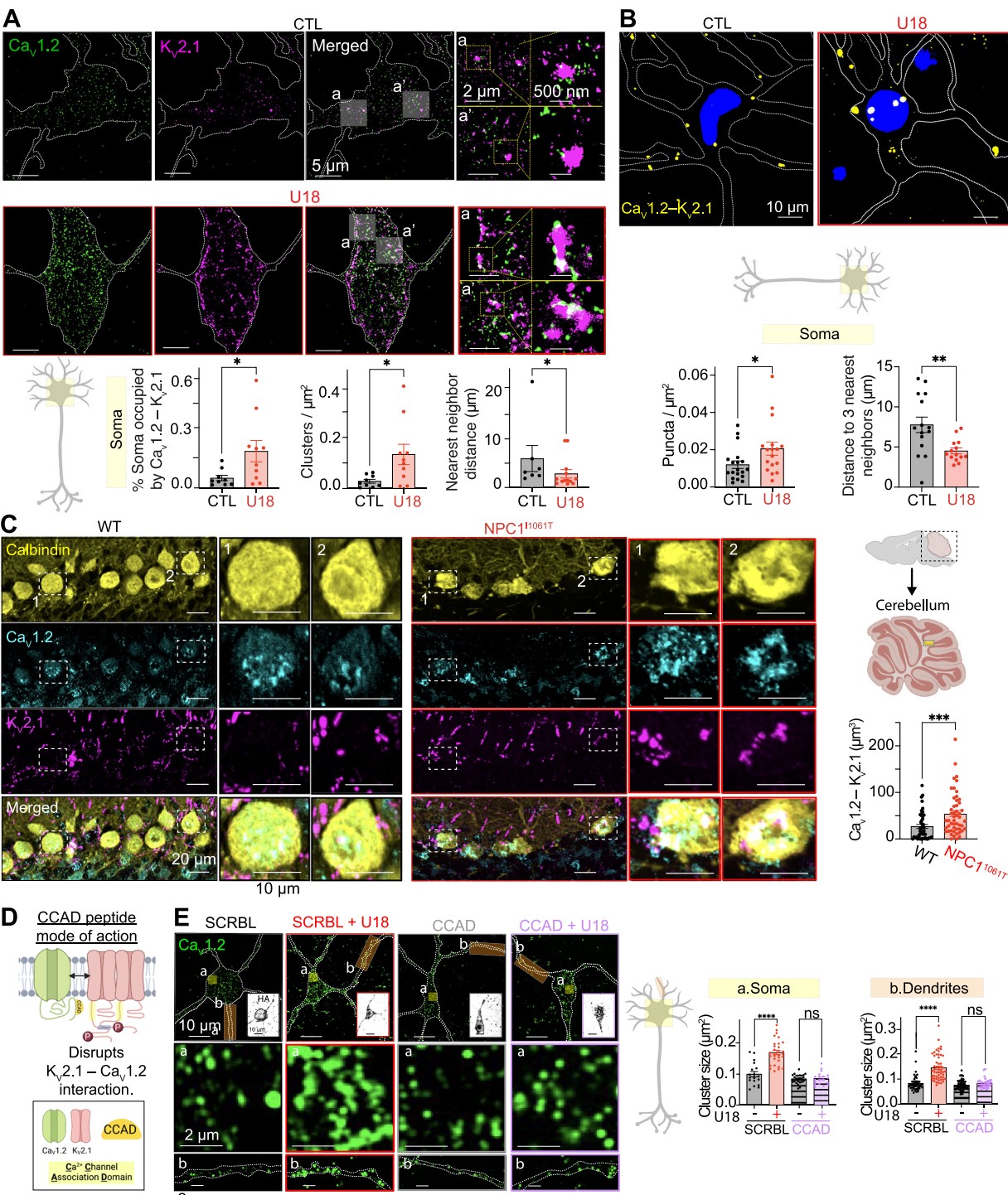

**Fig. 4 | NPC1 inhibition and NPC1 disease mutations increase spatial proximity between $K_V2.1$–$Ca_V1.2$. A** *Top*, representative super-resolution TIRF images of CTL (black) and U18-treated (red) neurons co-immunolabeled for $Ca_V1.2$ and $K_V2.1$. *Bottom*, quantification of % of soma occupied by $K_V2.1$–$Ca_V1.2$, $K_V2.1$–$Ca_V1.2$ cluster density and $K_V2.1$–$Ca_V1.2$ nearest cluster distance of CTL (black) and U18-treated (red) neurons in the soma region. $N = 9$ (CTL) and $n = 10$ (U18) neurons were analyzed across 2 independent isolations. **B** *Top*, representative $Ca_V1.2$ and $K_V2.1$ PLA images of CTL (black) and U18-treated (red) neurons. *Bottom*, quantification of PLA puncta density and nearest puncta distance of CTL (black) and U18-treated (red) neurons. $N = 19$ (CTL) and $n = 17$ (U18) neurons were analyzed across 2 independent isolations. **C** *Left*, representative maximum intensity projections from WT (black) and NPC1[I106IT] (red) cerebellar sagittal sections co-immunolabeled for Calbindin, $Ca_V1.2$ and $K_V2.1$. *Right*, quantification of $K_V2.1$–$Ca_V1.2$ colocalization volume in WT (black) and NPC1[I106IT] (red) neurons in the soma region. $n = 47$–48 (WT and NPC1[I106IT]) neurons

were analyzed across 3 animals. **D** Diagram detailing HA-TAT-$Ca^{2+}$ Channel Association Domain (CCAD) mode of action. **E** *Left*, representative super-resolution Airyscan images taken at a focal plane near the PM of CTL (black) and U18 (red) neurons incubated with the CCAD or HA-TAT-Scr scrambled peptide (SCRBL), and co-immunolabeled for $Ca_V1.2$ and HA. *Right*, quantification of PM $Ca_V1.2$ cluster size of CTL (black) and U18-treated (red) neurons co-incubated with CCAD or SCRBL peptide in the soma (left, yellow) and dendrite (right, orange) regions. $N = 18$ (SCRBL), $n = 30$ (SCRBL + U18), $n = 25$ (CCAD) and $n = 19$ (CCAD + U18) neurons and $n = 46$ (SCRBL), $n = 55$ (SCRBL + U18), $n = 57$ (CCAD) and $n = 34$ (CCAD + U18) dendrites were analyzed across 2 independent isolations. All error bars represent SEM. Statistical significance was calculated using Mann–Whitney (two-tail) and Unpaired $t$ tests (two-tail) in (**A**)–(**C**) and two-way ANOVA in (**E**); ns: not significant; *$P < 0.05$; **$P < 0.01$; ***$P < 0.001$; ****$P < 0.0001$. CTL is control, SCRBL is scramble, U18 is U18666A, and CCAD is calcium channel association domain.

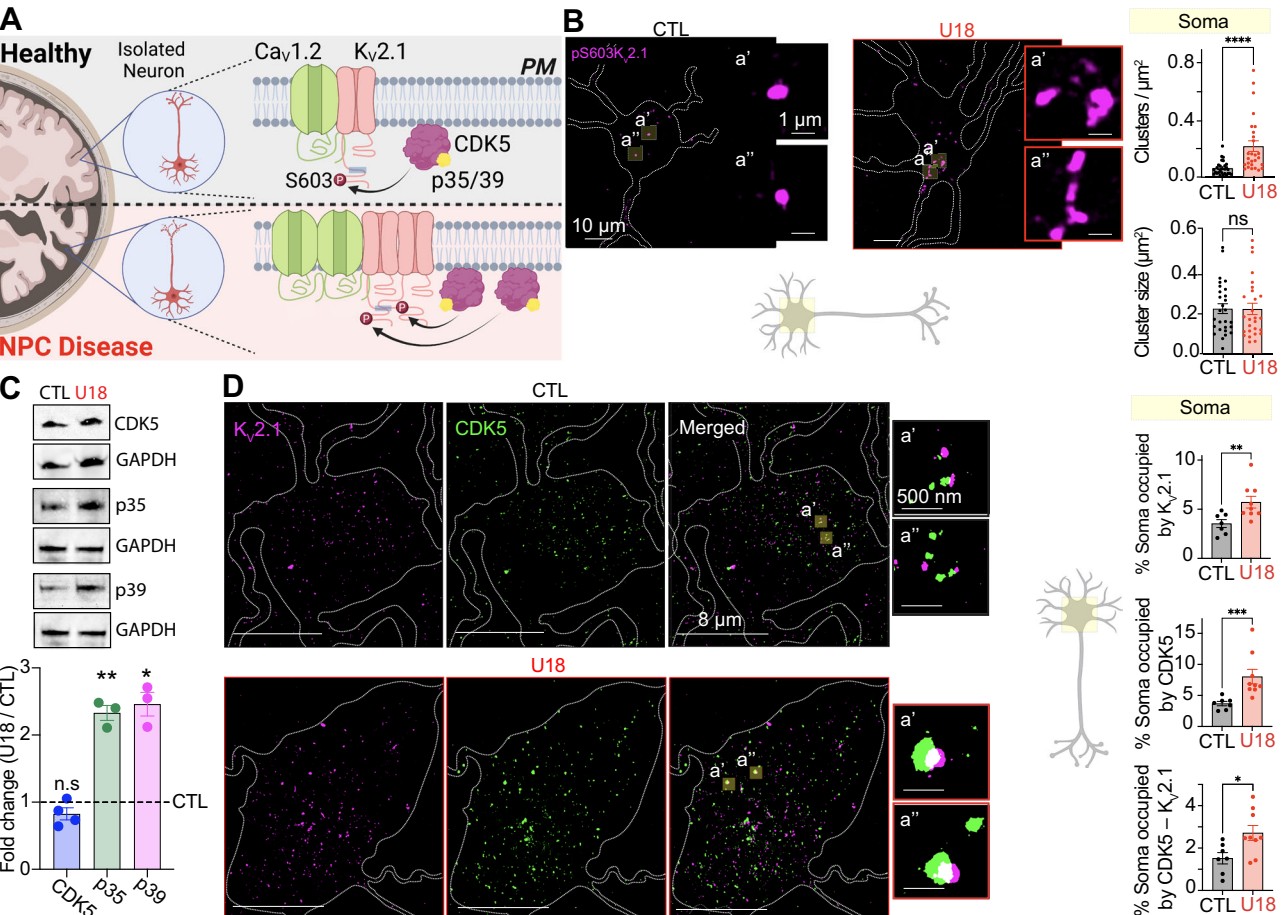

**Fig. 5 | Inhibition of NPC1 increases p35/p39 protein expression and CDK5 proximity to KV2.1. A** Schematic diagram of the hypothesis: CDK5 drives enhanced $K_V2.1$ phosphorylation in NPC disease. **B** *Left*, representative super-resolution Airyscan images taken at a focal plane near the PM of CTL (black) and U18-treated (red) neurons fixed and immunolabeled for $K_V2.1$p(S603). *Right*, quantification of PM $K_V2.1$(S603) cluster density and cluster size of CTL (black) and U18-treated (red) neurons in the soma region. $N = 28$ (CTL) and $n = 27$ (U18) neurons were analyzed across two independent isolations. **C** *Top*, representative blots of CDK5, p35, p39 with their respective GAPDH. *Bottom*, quantification of CDK5, p35 and p39 fold change (U18/CTL) normalized to GAPDH. $N = 4$ (CDK5), $n = 3$ (p35), and $n = 3$ (p39) across independent isolations. **D** *Left*, representative super-resolution TIRF images of CTL (black) and U18-treated (red) neurons co-immunolabeled for $K_V2.1$ and CDK5. *Right*, quantification of % of the soma occupied by $K_V2.1$, CDK5, and CDK5–$K_V2.1$. $N = 7$ (CTL) and $n = 9$ (U18) neurons were analyzed across 2 independent isolations. All error bars represent SEM. Statistical significance was calculated using Mann–Whitney (two-tail) and Unpaired $t$ test (two-tail) in (**B**) and (**D**), and one Sample $t$ test in (**C**); ns: not significant; $*P < 0.05$; $**P < 0.01$; $***P < 0.001$; $****P < 0.0001$. CTL is control, U18 is U18666A, CDK5 is Cyclin dependent kinase 5.

(Fig. S4C). To further elucidate if the physical proximity between $Ca_V1.2$ and $K_V2.1$ is altered in NPC1 disease conditions, we performed Proximity Ligation Assays (PLA). The PLA technique allows the detection and quantification of those $Ca_V1.2$ and $K_V2.1$ channels that are within 40 nm of one another[63]. Like super-res$_{TIRF}$ imaging, PLA experiments revealed an increase in the number of $Ca_V1.2$–$K_V2.1$ channel clusters within 40 nm of one another, with each hetero-cluster being closer to each other in the neuronal soma following loss of NPC1 function (Fig. 4B).

To corroborate that increased $Ca_V1.2$–$K_V2.1$ complexes occur in intact brain regions, we performed multiplexed immunolabeling on cerebellar tissue of WT and NPC1$^{I1061T}$ mutant mice (Fig. 4C). The cerebellum was chosen because Purkinje cells represent another highly vulnerable neuronal population in NPC1 disease[64,65]. Like isolated cortical neurons, $Ca_V1.2$ and $K_V2.1$ channels showed increased spatial proximity in mouse NPC1$^{I1061T}$ Purkinje neurons (Fig. 4C; movie S1). Thus, both NPC1 inhibition and disease mutations increase $Ca_V1.2$–$K_V2.1$ complexes.

$K_V2.1$ channels associate with $Ca_V1.2$ channels via a $Ca^{2+}$ Channel Association Domain (CCAD) located in the proximal cytoplasmic C-terminus of $K_V2.1$[2]. To test if CCAD-dependent interactions between

$K_V2.1$–$Ca_V1.2$ drive increases in $Ca_V1.2$ channel clusters following loss of NPC1 function we took advantage of a cell permeant synthetic peptide "HA-TAT-C1aB" (CCAD)[66] which selectively uncouples $K_V2.1$–$Ca_V1.2$ channel complexes[2]. Using this synthetic tool, and a corresponding peptide with a scrambled CCAD sequence "TAT-HA-C1aB-Scr" (SCRBL), we compared $Ca_V1.2$ distribution in isolated mouse (Fig. 4D, E and Fig. S4B) and rat (Fig. S5A) neurons under control and U18 conditions, as well as NPC1$^{-/-}$ cells (Fig. S4C). Analysis revealed that in all models increases in $Ca_V1.2$ density and cluster size were eliminated in the presence of CCAD, but not the Scr control peptide, providing evidence that U18 or NPC1$^{-/-}$ –associated elevations in the size of $Ca_V1.2$ clusters are dependent on interactions with $K_V2.1$ channels.

**CDK5-dependent phosphorylation of $K_V2.1$ drives NPC1-dependent increases in $Ca_V1.2$ clustering in isolated neurons**

Clustering of PM $K_V2.1$ channels is regulated by phosphorylation[32,67] with protein kinases like Cyclin Dependent Kinase 5 (CDK5) promoting $K_V2.1$ clustering[67] (Fig. 5A), and protein phosphatases like calcineurin de-clustering of $K_V2.1$[32]. To test if loss of NPC1 function increases phosphorylation of $K_V2.1$ channels we fixed and immunolabeled neurons with a phospho-specific antibody against the pS603 site of $K_V2.1$

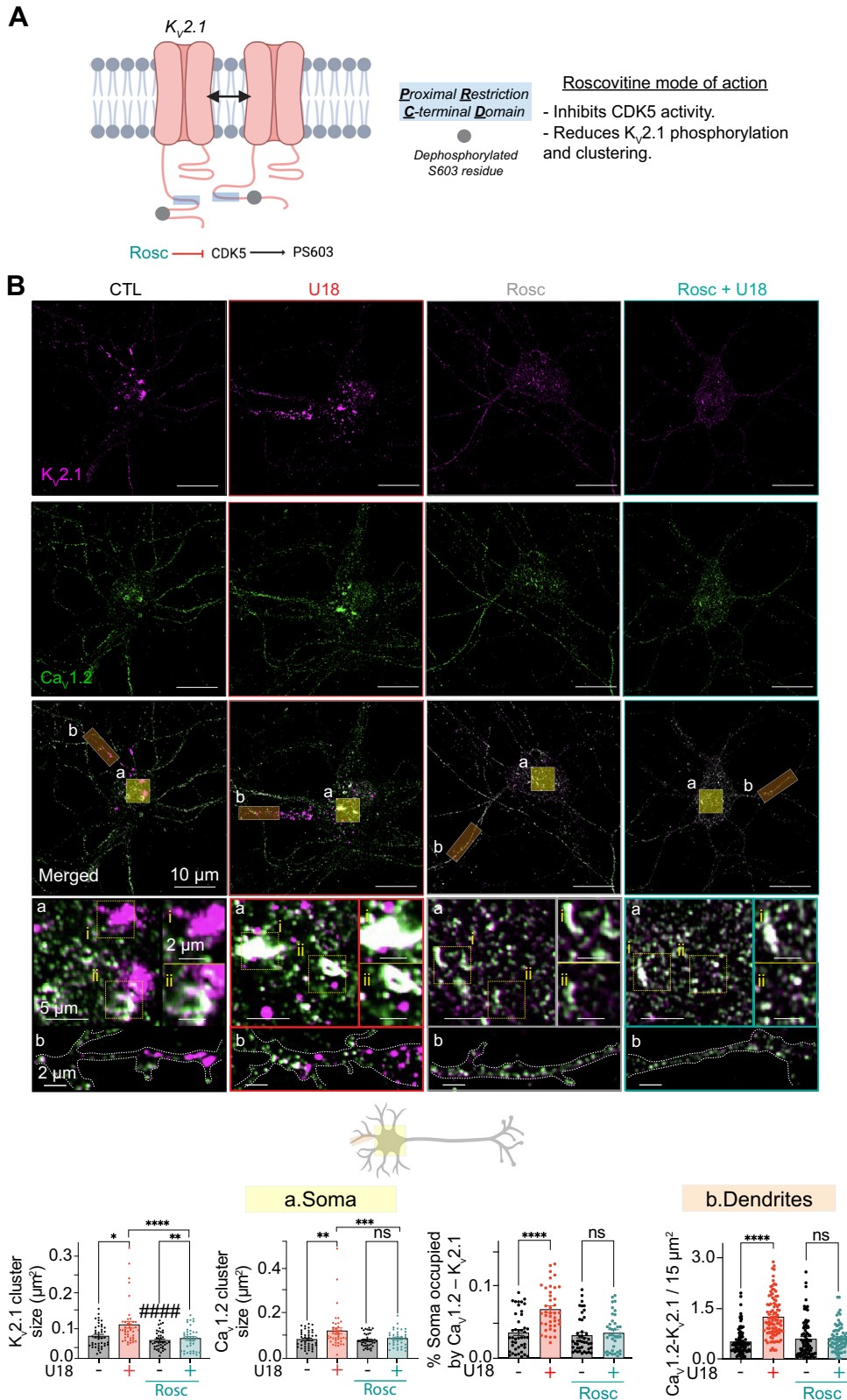

**Fig. 6 | NPC1-dependent increases in Ca$_V$1.2 are abrogated by inhibiting CDK5-dependent phosphorylation of K$_V$2.1. A** Schematic diagram detailing roscovitine (Rosc) mode of action. **B** *Top*, representative super-resolution Airyscan images taken at a focal plane near the PM of CTL (black) or U18-treated (red) neurons co-incubated with roscovitine (Rosc) (cyan) and co-immunolabeled for Ca$_V$1.2 and K$_V$2.1. *Bottom*, quantification of PM KV2.1 and Ca$_V$1.2 clustering size, and % of the soma occupied by Ca$_V$1.2–K$_V$2.1 (left, yellow) and Ca$_V$1.2–K$_V$2.1 area in the dendrite region (right, orange) of CTL (black), U18 (red) and Rosc (cyan) neurons. $N = 43$ (CTL), $n = 41$ (U18), $n = 43$ (Rosc) and $n = 38$ (Rosc+U18) neurons and $n = 89$ (CTL), $n = 94$ (U18), $n = 82$ (Rosc) and $n = 71$ (Rosc+U18) dendrites were analyzed across 5 independent isolations. All error bars represent SEM. Statistical significance was calculated using a two-way ANOVA test. ns: not significant; *$P < 0.05$; **$P < 0.01$; ***$P < 0.001$; ****$P < 0.0001$. #$P < 0.05$; ##$P < 0.01$; ####$P < 0.0001$. # indicates comparison with the CTL condition. CTL is control, U18 is U18666A, ROSC is Roscovitine.

which has been shown to be sensitive to CDK5 phosphorylation[67]. Subsequent super-res$_{TIRF}$ analysis revealed the density of somatic K$_V$2.1 pS603 clusters to be significantly increased in U18 treated neurons relative to control (Fig. 5B). To test if CDK5 underlies increased K$_V$2.1 phosphorylation in NPC disease we first measured CDK5 protein expression and found it was similar between control and U18 treated neurons (Fig. 5C). Next, we asked if CDK5 activating proteins p35/p39 have altered expression following loss of NPC1 function. Consistent with a report of increased CDK5 enzymatic activity in NPC disease models[68], expression of both p35 and p39 proteins was significantly elevated following inhibition of NPC1 (Fig. 5C). Interactions between CDK5 and p35 promote targeting and anchoring of the enzyme to the plasma membrane[69,70]; therefore, we performed dual-label super-res imaging for K$_V$2.1 and CDK5 and found the fractional amount of CDK5 overlapping with K$_V$2.1 increased when NPC1 was inhibited (Fig. 5D). Collectively, these experiments support a model whereby NPC1 loss-of-function increases p35/p39 expression which preferentially targets CDK5 to the plasma membrane. Here it is in closer proximity to K$_V$2.1 to increase phosphorylation-dependent clustering of K$_V$2.1 thereby promoting increased K$_V$2.1–Ca$_V$1.2 interactions.

### Targeting CDK5 rescues clustering phenotypes in NPC disease in vitro

To probe more directly that CDK5 activity drives phosphorylation of K$_V$2.1 to enhance K$_V$2.1–Ca$_V$1.2 complexes in models of NPC1 we conducted experiments using the small molecule CDK5 inhibitor, roscovitine (Fig. 6A). Cortical neurons were incubated with U18 and Roscovitine, before being fixed and immunolabeled for Ca$_V$1.2 and K$_V$2.1. In agreement with super-res analysis of K$_V$2.1 pS603 (Fig. 5B), U18 treatment significantly increased the area occupied by both Ca$_V$1.2 and K$_V$2.1 in somatic and proximal dendrites (Fig. 6B). In contrast, inhibiting CDK5 not only prevented the U18-dependent increases in both K$_V$2.1 and Ca$_V$1.2 clustering (Fig. 6B) but also abrogated U18-mediated elevations in Ca$_V$1.2–K$_V$2.1 hetero-cluster size and density (Fig. 6B). Interestingly, roscovitine did not alter steady-state K$_V$2.1–Ca$_V$1.2 hetero-cluster sizes in control neurons, suggesting a CDK5-independent mechanism for recruitment and binding of K$_V$2.1 to Ca$_V$1.2. This is perhaps not surprising given that several protein kinases can phosphorylate K$_V$2.1 at different positions and supports the concept that multiple protein kinases may tune K$_V$2.1–Ca$_V$1.2 interactions. As a final test, we began to probe molecular elements upstream of CDK5–p35/p39. To this end, we focused on mTORC1 as it has been shown to be hyperactive in NPC disease[15] and is proposed to inhibit AMPK[71] leading to increased CDK5 activity[72]. To test for an upstream role for mTORC1 we treated neurons with a combination of U18 (to inhibit NPC1) and/or Torin-1 (to inhibit mTORC1) before fixing and labeling for Ca$_V$1.2 and K$_V$2.1. Subsequent super-res imaging revealed that U18-dependent increases Ca$_V$1.2 and K$_V$2.1 clusters were sensitive to mTORC1 inhibition with K$_V$2.1–Ca$_V$1.2 nanocomplexes reduced back to control levels (Figure S5B). These data suggest that mTORC1 may be an upstream element controlling K$_V$2.1–Ca$_V$1.2 complexes in NPC disease. More experiments are required to determine the precise molecular pathway linking mTORC1 to alterations in K$_V$2.1–Ca$_V$1.2 complexes.

### K$_V$2.1-associated Ca$_V$1.2 influences the formation of ER–PM junctions

Phosphorylation of K$_V$2.1 within its Proximal Restriction and Clustering (PRC) domain generates a noncanonical FFAT (two phenylalanines (FF) in an Acidic Tract) motif that facilitates interactions with ER VAP proteins leading to the formation of ER–PM MCSs[34,35]. Given the increase in K$_V$2.1 clusters following loss of NPC1 function, we asked if ER–PM MCSs are also altered (Fig. 7A). To begin we performed a volumetric analysis using a live-neuron ER dye to test if basic ER morphology is altered following NPC1 inhibition and

determined that ER filament length and density, as well as ER branching density were the same in control and U18-treated neurons (Fig. S6). Therefore, NPC1 inhibition does not appear to result in gross changes to ER morphology. Next, we performed dual-label immunofluorescence against K$_V$2.1 and VAPA/B and determined using super-res$_{TIRF}$ imaging that VAPA/B cluster density, size, and overlap with K$_V$2.1 were all increased when NPC1 was inhibited (Fig. 7B). These data are consistent with increased ER–PM contact sites in NPC disease models. To test this hypothesis, we took advantage of two fluorescent tools: (i) SPLICS$_{S/L}$-P2A$^{ER-PM}$ [73] and (ii) MAPPER[74], which enable visualization and quantification of ER–PM MCSs in living neurons. The SPLICS$_{S/L}$-P2A$^{ER-PM}$ tool is based on the split-YFP protein fused to either an ER or PM target so it only emits fluorescence upon self-assembly at the ER–PM interface. A short or long spacer placed at the split-YFP ER targeted protein allows for the study of narrow (8–10 nm) or wide (40–50 nm) ER–PM MCSs[73]. Transfection of this ER–PM sensor into cortical neurons revealed a punctate distribution within either 40 nm or 10 nm of the TIRF footprint that represents the cortical ER in close apposition to the PM (Fig. 7C, D). Comparison of control and U18-treated neurons demonstrated that loss of NPC1 function resulted in an increase in the amount of ER in close proximity to the PM (Fig. 7C, D). As an alternative approach to quantify changes in ER–PM MCSs we used the MAPPER construct. The MAPPER tool is a GFP-tagged single peptide that contains the transmembrane domain of STIM1 (targeted to the ER) and a polybasic motif of the small G-protein Rit (PM binding)[74]. The polybasic motif interacts with negatively charged lipids at the cytoplasmic leaflet of the PM and enables visualization of ER–PM contacts between 10 and 25 nm. Like SPLICS$_{S/L}$-P2A$^{ER-PM}$, expression of MAPPER revealed a punctate distribution when visualized using super-res AiryScan microscopy at a focal plane close to the PM (Fig. 7E). Quantification of MAPPER images determined that MAPPER puncta density and size were both increased in U18-treated neurons at the soma and dendrites (Fig. 7E). Interestingly, when K$_V$2.1–Ca$_V$1.2 interactions were disrupted with the CCAD peptide, U18-dependent increases in ER–PM MCSs were partially reversed with decreases in the number and size of MAPPER puncta (Fig. 7E). These data suggest that K$_V$2.1-associated Ca$_V$1.2 plays an important role in generating ER–PM MCSs in NPC1 neurons.

### NPC1 inhibition remodels Ca$^{2+}$ handling proteins at ER–PM MCSs to potentiate Ca$^{2+}$ signaling in vitro

At somatic ER–PM MCSs, in addition to Ca$_V$1.2, K$_V$2.1 is in close spatial proximity with several other Ca$^{2+}$ handling proteins, including Ryanodine Receptors (RyR) and Sacro/Endoplasmic Reticulum Ca$^{2+}$-ATPase (SERCA)[31,75]. Considering the increases in K$_V$2.1–Ca$_V$1.2 complexes (Fig. 4) and ER–PM MCSs (Fig. 7) observed following the loss of NPC1 function we wanted to determine if RyR and SERCA are enriched at these ER–PM contacts (Fig. 8A). To test if SERCA distribution is altered at K$_V$2.1-forming nanodomains we performed super-res$_{TIRF}$ on neurons fixed and co-immunolabeled for K$_V$2.1 and SERCA. In control and U18-treated neurons SERCA was observed in the TIRF footprint, indicating close spatial proximity to the PM (Fig. S7A). Furthermore, U18 treatment significantly enhanced SERCA cluster area and overlap with K$_V$2.1 (Figure S7A). These data align with previously published results that SERCA expression is significantly increased in NPC disease models[23]. To assess RyR distribution relative to K$_V$2.1–Ca$_V$1.2 domains, we co-immunolabeled for Ca$_V$1.2 and RyR with or without U18 treatment to inhibit NPC1. Using super-res AiryScan confocal microscopy we quantified RyR and Ca$_V$1.2 distribution at a focal plane close to the PM and found that in addition to expected increases in Ca$_V$1.2 cluster size, their proximity to one another increased when NPC1 function is lost (Fig. 8B). On average, there was a 40 % increase in overlapping Ca$_V$1.2–RyR pixels in the soma and a doubling of overlapping pixels in the dendrites of NPC loss of function neurons relative to control. These

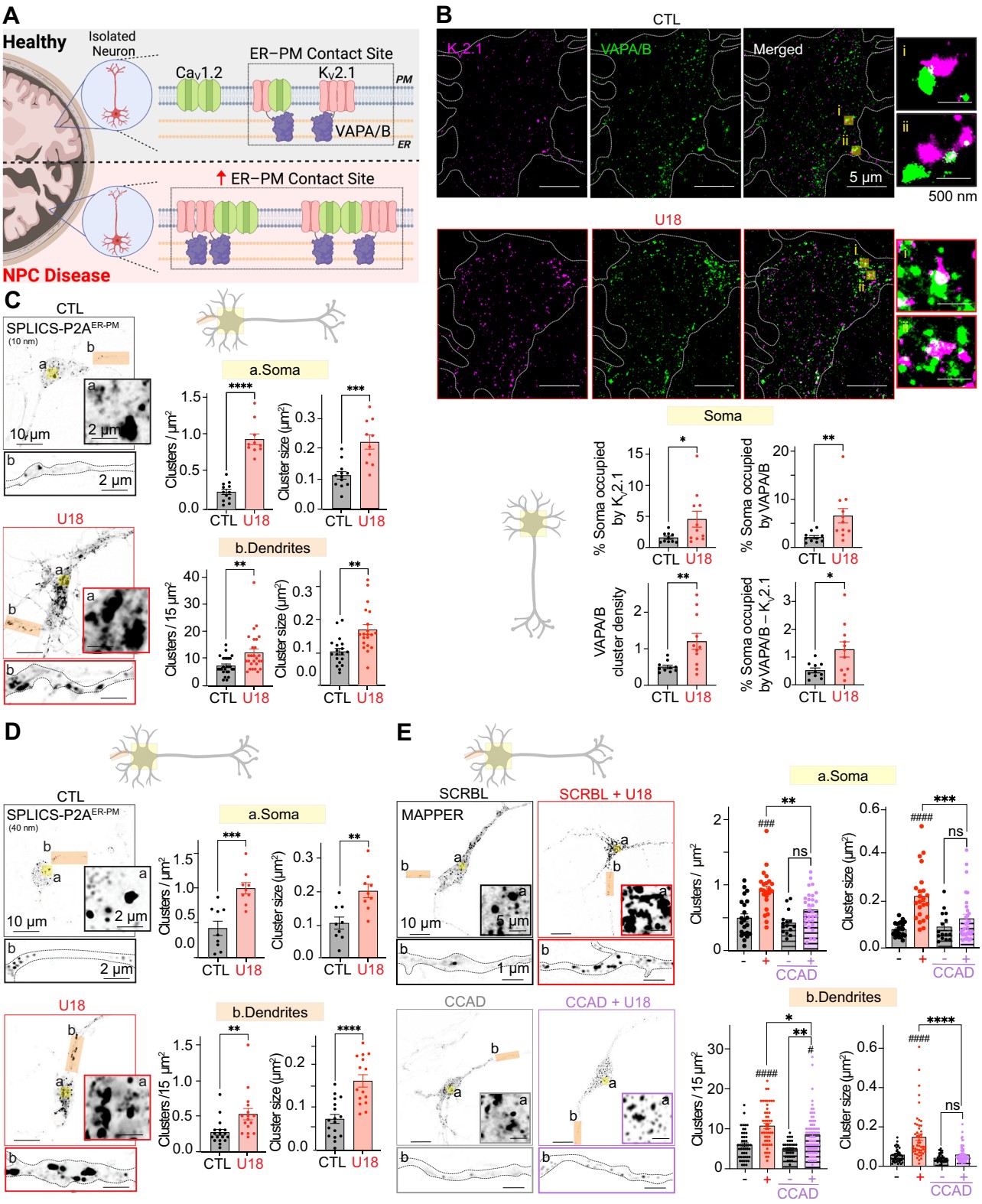

findings reveal that both RyR and SERCA are enriched at $K_V2.1$ associated ER–PM MCSs following the loss of NPC1 function.

Considering the importance of $K_V2.1$ for tuning ER–PM $Ca^{2+}$ nanodomains[2,31] and the significant reorganization of $K_V2.1$–$Ca_V1.2$–RyR–SERCA that occurs following loss of NPC1 function, we next wanted to understand if this molecular remodeling results in augmented $Ca^{2+}$ signaling at these $K_V2.1$ associated ER–PM contacts. To monitor and quantify $Ca^{2+}$ signaling at $K_V2.1$ ER–PM MCSs we took

advantage of a recently described GCaMP $Ca^{2+}$ sensor, GCaMP3-$K_V2.1_{P4O4W}$ which is GCaMP3 appended to the N-terminus of non-conducting $K_V2.1$[31]. Transfection of GCaMP3-$K_V2.1_{P4O4W}$ into cortical neurons resulted in its distinct localization to ER–PM MCSs (Fig. 8C). Using TIRF microscopy to visualize GCaMP3-$K_V2.1_{P4O4W}$ in close apposition to the PM we quantified spontaneous changes in GCaMP3 intensity from neurons expressing low amounts of the $Ca^{2+}$ biosensor. Analysis of GCaMP3s signals determined that in NPC loss of function

**Fig. 7 | NPC1 regulates the size and density of $K_V2.1$–VAPA/B and ER–PM MCSs.**
**A** Schematic diagram of the hypothesis: loss of NPC1 function leads to increased ER-PM junctions. **B** *Top*, representative super-resolution TIRF images of CTL (black) and U18-treated (red) neurons co-immunolabeled for $K_V2.1$ and VAPA/B. *Bottom*, quantification of % of the soma occupied by $K_V2.1$, VAPA/B, and $K_V2.1$–VAPA/B as well as VAPA/B cluster density. $N = 10$ (CTL) and $n = 11$ (U18) neurons were analyzed across 2 independent isolations. **C** *Left*, representative images taken at a focal plane near the PM of CTL (black) and U18 (red) neurons transfected with $SPLICS_S$-$P2A^{ER-PM}$ probe (10 nm). *Right*, quantification of $SPLICS_L$-$P2A^{ER-PM}$ cluster density and size of CTL (black) and U18 (red) neurons in the soma (top, yellow) and dendrite (bottom, orange) regions. $N = 9$ (CTL) and $n = 9$ (U18) neurons and $n = 17$ (CTL) and $n = 15$ (U18) dendrites were analyzed across one isolation. **D** Same as (**C**), only transfected with $SPLICS_L$-$P2A^{ER-PM}$ probe (40 nm). $N = 12$ (CTL) and $n = 10$ (U18) neurons and

$n = 19$ (CTL) and $n = 20$ (U18) dendrites were analyzed across one isolation. **E** *Left*, representative images taken at a focal plane near the PM of MAPPER transfected CTL (black), U18 (red) and CCAD + U18 (purple) neurons. Bottom, quantification of MAPPER puncta density and puncta size of CTL (black), U18 (red) and CCAD + U18 (purple) neurons in the soma (top, yellow) and dendrite (bottom, orange) regions. $N = 22$ (CTL), $n = 23$ (U18), $n = 15$ (CCAD) and $n = 30$ (CCAD + U18) neurons and $n = 47$ (CTL), $n = 50$ (U18), $n = 34$ (CCAD) and $n = 72$ (CCAD + U18) dendrites were analyzed across 3 independent isolations. All error bars represent SEM. Statistical significance was calculated using Mann–Whitney (two-tail) and unpaired $t$ test (two-tail) in (**B**)–(**D**) and two-way ANOVA in (**E**). ns: not significant; $*P < 0.05$; $**P < 0.01$; $***P < 0.001$; $****P < 0.0001$. $^\#P < 0.05$; $^{\#\#}P < 0.01$; $^{\#\#\#\#}P < 0.0001$. $^\#$ indicates comparison with the CTL condition. CTL is control, SCRBL is scramble, U18 is U18666A, and CCAD is calcium channel association domain.

neurons, the amplitude of spontaneous $Ca^{2+}$ signals at $K_V2.1$-forming ER–PM MCSs effectively doubled compared to control (Fig. 8C). To test the role of $Ca_V1.2$ channels in facilitating this increase we uncoupled $K_V2$–$Ca_V1$ interactions at ER–PM MCS (co-incubation with the CCAD peptide) which resulted in neurons being refractory to U18 and having significantly decreased responses relative to control (Fig. 8C; Movie S2). Similar results were observed when neurons were treated with a cell-penetrating FFAT peptide to uncouple $K_V2.1$ from VAPA/B. Finally, to test if CCAD decreased $Ca^{2+}$ elevations near $K_V2.1$ due to selective uncoupling of $K_V2.1$–$Ca_V1.2$ or due to hyperpolarization in the membrane potential we performed experiments using a genetically encoded voltage indicator (ArchLight-Q239[76]). High-speed imaging of ArchLight-Q239 revealed that U18-treated neurons had similar fluctuations in intensity when compared to U18 and CCAD (Fig. S7B), suggesting that CCAD-dependent decreases in $Ca^{2+}$ (Fig. 8D) occur through selective uncoupling of $K_V2.1$–$Ca_V1.2$ rather than alterations in neuronal excitability.

Taken together, these results demonstrate a role for NPC1 in the downstream organization of spatial and functional coupling between PM $K_V2.1$–$Ca_V1.2$ and the ER to regulate ER–PM $Ca^{2+}$ nanodomains.

## Remodeling of ER–PM and ER–Mito MCSs results in neurotoxic increases in mitochondrial $Ca^{2+}$ and neurodegeneration

Our data suggest there are significant alterations to global and local (ER–PM MCSs) $Ca^{2+}$ nanodomains in NPC neurons. Considering the importance of appropriately regulated $Ca^{2+}$ levels for neuronal signaling, we next wanted to determine if these excessive $Ca^{2+}$ events had toxic consequences for neuronal viability. In neurons, the ER and mitochondria act as major buffers in response to high cytosolic $Ca^{2+}$ levels. Functional coupling between the $IP_3R$ and the Voltage-Dependent Anion channel 1 (VDAC1) at ER–Mito MCSs ensures a steep local $Ca^{2+}$ gradient for $Ca^{2+}$ to move into mitochondria to maintain energy production and cell health[77,78]. However, deviant mitochondrial $Ca^{2+}$ ($Ca^{2+}_{Mito}$) levels can lead to altered bioenergetics, ATP production, and Reactive Oxygen Species (ROS) production, triggering necrosis or apoptosis[79–81]. Therefore, we tested the hypothesis that NPC1-dependent elevations in cytoplasmic $Ca^{2+}$ increases $Ca^{2+}_{Mito}$ and leads to neurotoxicity (Fig. 9A).

We began by characterizing the gross structural organization of the ER and mitochondria in control and U18-treated cells loaded with mitochondria- and ER-permeable dyes. Super-res AiryScan microscopy was performed to measure the overlapping signal near the PM. These analyses demonstrated that the proportion of ER overlapping with mitochondria was very similar in both conditions (Fig. S8A), suggesting that at this resolution there are no obvious rearrangements of these organelle membranes. Next, we asked more specifically if ER–Mito $Ca^{2+}$ domains were affected by NPC1 function. At ER–Mito MCSs the $IP_3R$ and VDAC1 are coupled through interactions with the chaperone protein GRP75[82]. To determine if the $IP_3R$–GRP75–VDAC1 signaling axis, which has been implicated in the progression of other neurodegenerative diseases like Alzheimer's,

Parkinson's, and Amyotrophic Lateral Sclerosis, undergoes reorganization in NPC disease we took a two-pronged approach leveraging $NPC1^{-/-}$ cells and neurons treated with U18. First, using control and $NPC1^{-/-}$ cells, or vehicle and U18-treated neurons, we immunolabeled for the GRP75 protein and acquired super-resolution AiryScan confocal images. Fig. 9B shows that the total fluorescence intensity, cluster density and cluster size was augmented in U18-treated neurons compared to control. Similarly, $NPC1^{-/-}$ cells also showed higher GPR75 intensity levels and larger clusters (Fig. S8B), indicating enhanced GPR75 at ER–Mito MCSs.

To understand if increased GRP75 cluster size and intensity also reflects remodeling of $IP_3R$ and VDAC1 proteins at ER–Mito MCSs we quantified the overlap between these proteins using two models of NPC disease. First, using neurons, we fixed and co-immunolabeled for VDAC1 and $IP_3R1$ before imaging using super-resolution Airyscan confocal microscopy. As shown in Fig. 9C, analyses of resultant 2-color images determined that the proportion, density, and size of VDAC1–$IP_3R$ overlapping clusters were all significantly increased in U18 conditions. Similar experiments performed using $NPC1^{-/-}$ cells (Fig. S8C), HEK cells that have endogenous $IP_3R1$ tagged with eGFP (Fig. S8D[83]), and blinded neuronal experiments (Fig. S8E) revealed that NPC1 knockout or inhibition increased the amount of overlapping $IP_3R$–VDAC1. Taken together, our data indicated that the $IP_3R$–GPR75–VDAC1 signaling axis is upregulated following NPC1 loss-of-function, suggesting $Ca^{2+}$ flux from ER to mitochondria may be altered in NPC disease.

The remodeling of ER–Mito $Ca^{2+}$ handling proteins following loss of NPC1 function prompted us to ask if free $Ca^{2+}$ levels within the ER and mitochondria are also altered in neurons. Previously it has been shown in non-excitable cell models of NPC disease, including cells harboring the most prevalent patient mutation, that loss of NPC1 function results in decreased ER $Ca^{2+}$ ($Ca^{2+}_{ER}$) levels due to increased ER $Ca^{2+}$ leak through $IP_3R1$[17,23]. To test if similar results occur in neurons, we transfected neurons with the ER-targeted GCaMP indicator, GCaMP6-150[84], and treated with vehicle control or U18. Aligned with published results, loss of NPC1 function resulted in about a 50 % decrease in GCaMP intensity (Fig. 10A). These data support published data that loss of NPC1 function decreases $Ca^{2+}_{ER}$.

In NPC disease, decreases in $Ca^{2+}_{ER}$ occur due to enhanced opening of $IP_3R1$ receptors[17]. Given the enhanced proximity and size of $IP_3R1$–VDAC1 interactions (Fig. 9C), we tested if leaky $IP_3R1$ receptors result in elevated neuronal mitochondrial $Ca^{2+}$ ($Ca^{2+}_{Mito}$). To evaluate $Ca^{2+}_{Mito}$ we transfected cortical neurons with a mitochondria-targeted RCaMP plasmid (pCAG-mito-RCaMP1h)[85], and incubated neurons with vehicle or U18 for 24 h. Consistent with the model that leaky $IP_3R1$ on ER membranes drive $Ca^{2+}$ entry into mitochondria, normalized pCAG-mito-RCaMP1h intensities were significantly elevated following the loss of NPC1 function (Fig. 10B). To test a central role for ER $Ca^{2+}$ stores and $IP_3R1$ in facilitating increases in neuronal $Ca^{2+}_{Mito}$ we treated with vehicle or xestospongin C (XestoC; $IP_3R$ inhibitor) and found that U18-dependent elevations in $Ca^{2+}_{Mito}$ were abrogated (Fig. 10B). To further

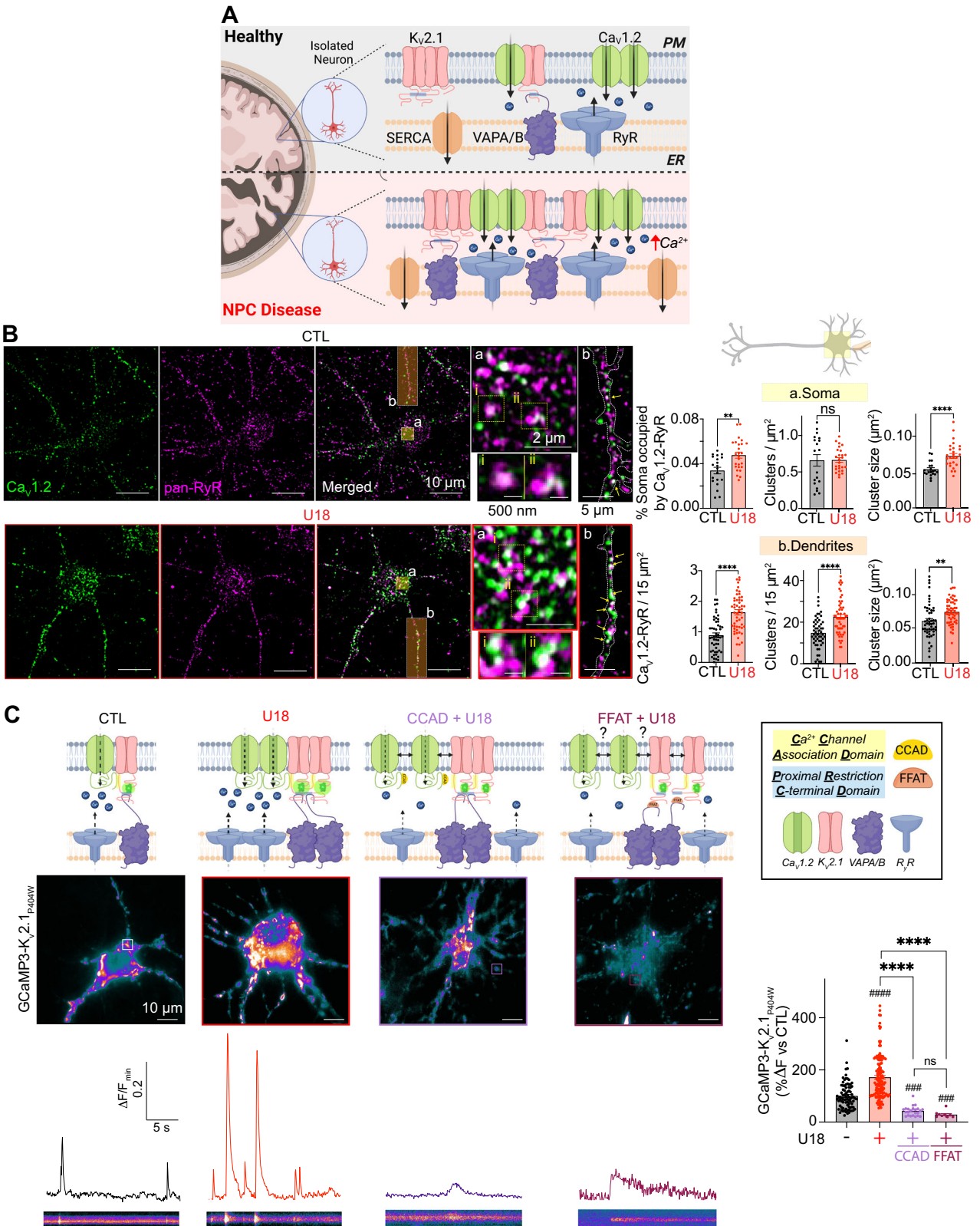

define the upstream source of Ca²⁺, we treated neurons with vehicle control and nifedipine (L-type Ca$_V$1 inhibitor; Nif). Consistent with the model that Ca$_V$1 channels play a central role in mediating increases in Ca²⁺$_{Mito}$ in NPC disease, treatment with nifedipine abolished increases in Ca²⁺$_{Mito}$ (Fig. 10B). Interestingly, AP blockage by TTX also prevented elevated Ca²⁺$_{Mito}$ in U18 neurons (Fig. 10A), suggesting that increased neuronal excitability observed in NPC disease leads to elevated

Ca²⁺$_{Mito}$. Since increased K$_V$2.1–Ca$_V$1.2 interactions play a major role in Ca$_V$1.2 clustering (Fig. 4) and in localized Ca²⁺ activity at K$_V$2.1-forming ER−PM MCSs (Fig. 8), we predicted that K$_V$2-dependent interactions with Ca$_V$1 may underlie elevations in Ca²⁺$_{Mito}$ in NPC disease. To test this prediction, we incubated neurons with CCAD to uncouple K$_V$2 from Ca$_V$1 and determined that like nifedipine treatment, disengaging K$_V$2 from Ca$_V$1 resulted in neuronal Ca²⁺$_{Mito}$ being refractory to U18

**Fig. 8 | Loss of NPC1 function spatially and functionally remodels RyR–Ca$_V$1.2–K$_V$2.1 signaling nanodomains. A** Schematic diagram of the hypothesis: NPC1 deficient neurons have ER–PM domains enriched in RyR–Ca$_V$1.2–K$_V$2.1–SERCA and increased spontaneous Ca$^{2+}$ activity at K$_V$2.1-associated ER–PM MCSs. **B** *Left*, representative super-resolution Airyscan images taken at a focal plane near the PM of CTL (black) and U18-teated (red) neurons co-immunolabeled for Ca$_V$1.2 and RyR. *Right*, quantification of % soma occupied by Ca$_V$1.2–RyR, Ca$_V$1.2–RyR cluster density and Ca$_V$1.2–RyR cluster size of CTL (black) and U18-treated (red) neurons in the soma (top, yellow) and dendrite (bottom, orange) region. $N = 20$–21 (CTL) and $n = 26$ (U18) neurons and $n = 52$–53 (CTL) and $n = 54$–55 (U18) dendrites were analyzed across 2 independendent isolations. **C** *Top*, schematic diagram of hypothesis: NPC1 deficient neurons (U18, red) have increased Ca$^{2+}$ activity and K$_V$2.1-associated Ca$_V$1.2, while disrupting K$_V$2.1–Ca$_V$1.2 or K$_V$2.1–VAPA/B interactions (CCAD, purple or FFAT, red-brown) abrogates such effects. *Middle left*,

representative super-resolution TIRF images of neurons transfected with GCaMP3-Kv2.1$_{P4O4W}$. *Bottom left*, intensity time series and kymographs of spontaneous activity taken from the square region of interest. *Middle right*, quantification of GCaMP3-Kv2.1$_{P4O4W}$ peak amplitude of CTL (black), U18 (red), CCAD + U18 (purple) and FFAT + U18 (red-brown) neurons. $N = 85$ (CTL), $n = 141$ (U18), $n = 23$ (CCAD + U18) and $n = 10$ (FFAT + U18) GCaMP3-Kv2.1$_{P4O4W}$ peaks were analyzed across 3 independent isolations. All error bars represent SEM. Statistical significance was calculated using the following tests: Unpaired (two-tail) and Mann–Whitney $t$ tests (two-tail) in (**B**) and Kruskal–Wallis test in (**C**). ns: not significant; *$P < 0.05$; **$P < 0.01$; ***$P < 0.001$; ****$P < 0.0001$. #$P < 0.05$; ##$P < 0.01$; ####$P < 0.0001$. # indicates comparison with the CTL condition. CTL is control, SCRBL is scramble, U18 is U18666A, CCAD is calcium channel association domain, and FFAT is two phenylalanines (FF) in an acidic tract.

(Fig. 10C). Similar results were observed with roscovitine (Fig. S9B) and place K$_V$2.1–Ca$_V$1 interactions at ER–PM MCSs as the most upstream source of Ca$^{2+}$ driving increases in Ca$^{2+}_{Mito}$ in NPC disease. Alterations in Ca$^{2+}_{Mito}$ are often accompanied by alterations in reactive oxygen species (ROS) and changes in mitochondrial membrane potential (MMP) therefore we tested if they were also altered in models of NPC. Like Ca$^{2+}_{Mito}$, both ROS and MMP were differentially changed following NPC1 inhibition with levels of ROS increased (Fig. 10D) and MMP less polarized (Fig. 10E). Significantly, treatment of neurons with CCAD rescued defects in ROS and MMP.

Sustained elevations in Ca$^{2+}_{Mito}$ or alterations in ROS and MMP can be detrimental to neuronal health and are frequently triggers for neurodegeneration. Thus, we tested if the mitochondrial phenotypes noted above correlate with changes in neuronal fidelity in NPC disease. As shown in Fig. 10F, viability was significantly decreased following 48 h of NPC1 inhibition. Treatment with XestoC or roscovitine abolished NPC1-driven toxicity (Fig. 10F) emphasizing the important roles IP$_3$R and CDK5 play in decreasing neuron viability. Moreover, uncoupling K$_V$2.1–Ca$_V$1.2 interactions through the CCAD peptide completely rescued neurons from toxicity (Fig. 10F).

Collectively, these data suggest that augmented Ca$_V$1.2–K$_V$2.1 interactions at ER–PM MCSs drive deviant increases in Ca$^{2+}_{Mito}$ to promote neurotoxicity in NPC disease.

## Discussion

In the present study, we provide evidence linking the NPC1 cholesterol transport to the regulation of ER–PM and ER–Mito Ca$^{2+}$ signaling nanodomains and neurodegeneration. We show in models of NPC disease that loss of NPC1 cholesterol transporter function facilitates a K$_V$2.1-dependent reorganization of Ca$_V$1.2–SERCA–RyR Ca$^{2+}$ handling proteins to potentiate Ca$^{2+}$ entry at ER–PM MCSs. In parallel with these changes, we also report the molecular remodeling of the IP$_3$R1–GRP75–VDAC1 Ca$^{2+}$ complexes at ER–Mito MCSs. Collectively, NPC1-dependent remodeling of these junctions creates a damaging feed-forward Ca$^{2+}$ signaling axis which drives Ca$^{2+}$ entry into mitochondria leading to neurotoxicity. Importantly, we show that uncoupling K$_V$2–Ca$_V$1 interactions at ER–PM MCSs rescues the NPC1-dependent mitochondrial Ca$^{2+}$ defects and neurotoxicity, strongly suggesting that Ca$^{2+}$ entry at upstream ER–PM MCSs is a key driver of Ca$^{2+}$-dependent neurodegeneration (see Fig. S10 for model).

NPC disease is a neurodegenerative disorder that is characterized by the accumulation of cholesterol within the lysosome. In addition to this cellular phenotype, neurons from several models of NPC disease are hyperexcitable and fire APs at a higher frequency than WT neurons[52]. A simple prediction was that this hyperexcitability phenotype would increase voltage-dependent Ca$^{2+}$ entry into NPC neurons. Indeed, we demonstrate that NPC1 loss of function leads to more frequent, electrically driven oscillations in intracellular Ca$^{2+}$. In addition, we have also discovered that alterations in the nanoscale organization of Ca$^{2+}$ handling proteins, localized at ER–PM MCSs, further potentiates Ca$^{2+}$

entry. Central to these structural rearrangements is K$_V$2.1 which influences the distribution and activity of Ca$_V$1 channels, RyR, and SERCA[2,31,36,75]. We find that increased clustering of K$_V$2.1 enhances interactions with Ca$_V$1 channels to increase their density and cluster size. This is important as the distribution of Ca$_V$1 channels dictates their function with clustered Ca$_V$1 channels functionally communicate with one another via their C-termini to mediate their cooperative gating[57–61]. The opening of Ca$_V$1 channels within a cluster is driven by the channel with the highest open probability leading to an overall enhancement in channel activity and resultant amplification of Ca$^{2+}$ influx. Observations of modified Ca$_V$1 channel nanoscale structure–activity relationships have been proposed as key contributors to the disease progression of pathologies such as Timothy syndrome[60], hypertension[60], and diabetes[86], and here we detail similar observations in a neurodegenerative disorder. It is conceivable that similar changes in the nanoscale distribution and activity of Ca$_V$1 channels may occur in other neurodegenerative disorders where dysfunctional Ca$^{2+}$ signaling and altered cholesterol homeostasis have been observed[8,9,38,45,87–89].

The molecular link between NPC1 and enhanced K$_V$2.1–Ca$_V$1 interactions appears to involve the protein kinase CDK5. Phosphorylation of K$_V$2.1 by CDK5 enhances channel clustering[67]. Furthermore, CDK5 kinase activity is elevated in NPC disease[68,90]. We find that CDK5 activators, p35/p39 have increased expression in NPC disease and inhibition of CDK5 kinase activity with roscovitine abrogates enhancement of both K$_V$2.1 and Ca$_V$1.2 clustering in the PM of NPC neurons. Based on this information, we propose a model wherein loss of NPC function increases CDK5 partitioning to the plasma membrane resulting in a shift in the fractional amount of phosphorylated K$_V$2.1, enhancing its clustering and consequently the distribution and activity of Ca$_V$1 channels at the PM. Interestingly, a parallel consequence of increased phosphorylation of K$_V$2.1 would be a decrease in its overall activity as a K$^+$ conducting channel[91] which may potentially contribute to the hyperexcitability phenotype in NPC[52]. Many questions arise from our findings, for example, what is the link between NPC1 and CDK5 activity? We provide evidence that hyperactive mTORC1 activity may be part of the molecular pathway between NPC1 and increased ER–PM Ca$^{2+}$ activity in NPC but more experiments are required to understand the precise steps that allow such signaling to proceed. Second, with the increases in excitability and Ca$^{2+}$ entry: why doesn't calcineurin activity act as a homeostatic counterbalance to CDK5 to ensure K$_V$2.1 clustering at ER–PM contact sites stays in a 'physiological' range? Perhaps the depleted lysosomal Ca$^{2+}$ stores in NPC depress the catalytic activity of calcineurin[92,93]. We provide evidence that CDK5 and K$_V$2.1 come into closer proximity following NPC1 inhibition which perhaps favors a net increase in K$_V$2.1 phosphorylation and consequently K$_V$2.1–Ca$_V$1 complex formation. Future analyses should begin targeting these key questions to unmask and define the complete molecular cascade that exists between NPC1 cholesterol efflux and the tuning of ion channel distribution at ER–PM MCSs.

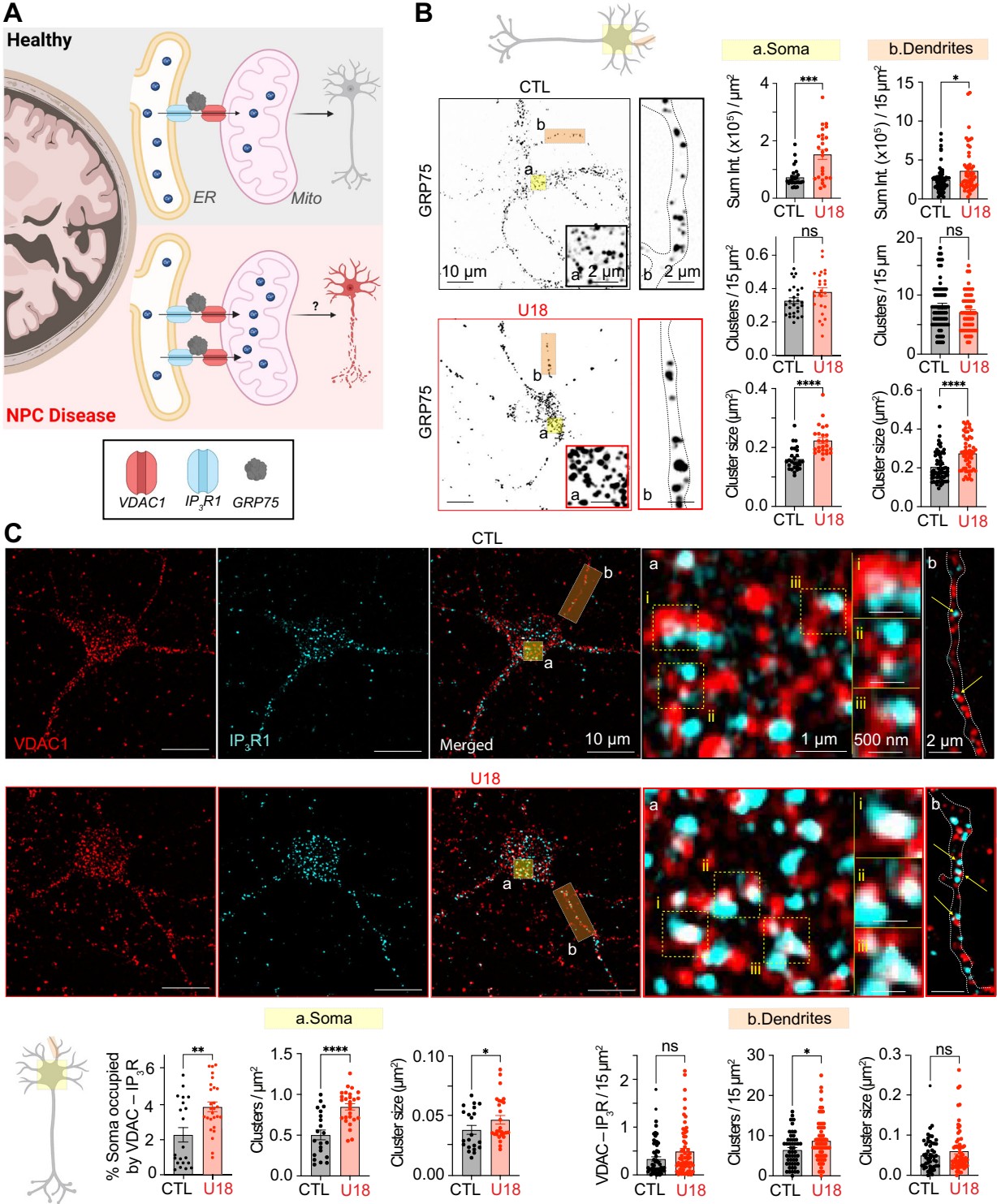

**Fig. 9 | NPC1 dysfunction increases GRP75 and IP₃R1–VDAC1 clustering.**
**A** Schematic diagram of the hypothesis: Enhanced IP₃R–GRP75–VDAC signaling axis in NPC neurons leads to an aberrantly increased mitochondrial Ca²⁺ and neurotoxicity. **B** Left, representative super-resolution Airyscan images of CTL (black) and U18-treated neurons immunolabeled for GRP75. *Right*, quantification of GRP75 total intensity, cluster density and cluster size of CTL (black) and U18 (red) neurons in the soma (left, yellow) and dendrite (right, orange) regions. *N* = 27 (CTL) and *n* = 25 (U18) neurons and *n* = 65 (CTL) and *n* = 54–55 (U18) dendrites were analyzed across 2 independent isolations. **C** *Top*, representative images of CTL (black) and

U18 (red) neurons fixed and immunolabeled for IP₃R and VDAC1. *Bottom*: quantification of % soma occupied by IP₃R–VDAC1, IP₃R–VDAC1 cluster density and IP₃R–VDAC1 cluster size of CTL (black) and U18-treated (red) neurons in the soma (left, yellow) and dendrite (right, orange) regions. *N* = 21 (CTL) and *n* = 27 (U18) neurons and *n* = 48 (CTL) and *n* = 65 (U18) dendrites across 2 independent isolations. All error bars represent SEM. Statistical significance was calculated using Mann–Whitney (two-tail) and Unpaired *t* tests (two-tail) in (**B**) and (**C**). ns: not significant; *P < 0.05; **P < 0.01; ***P < 0.001; ****P < 0.0001. CTL is control and U18 is U18666A.

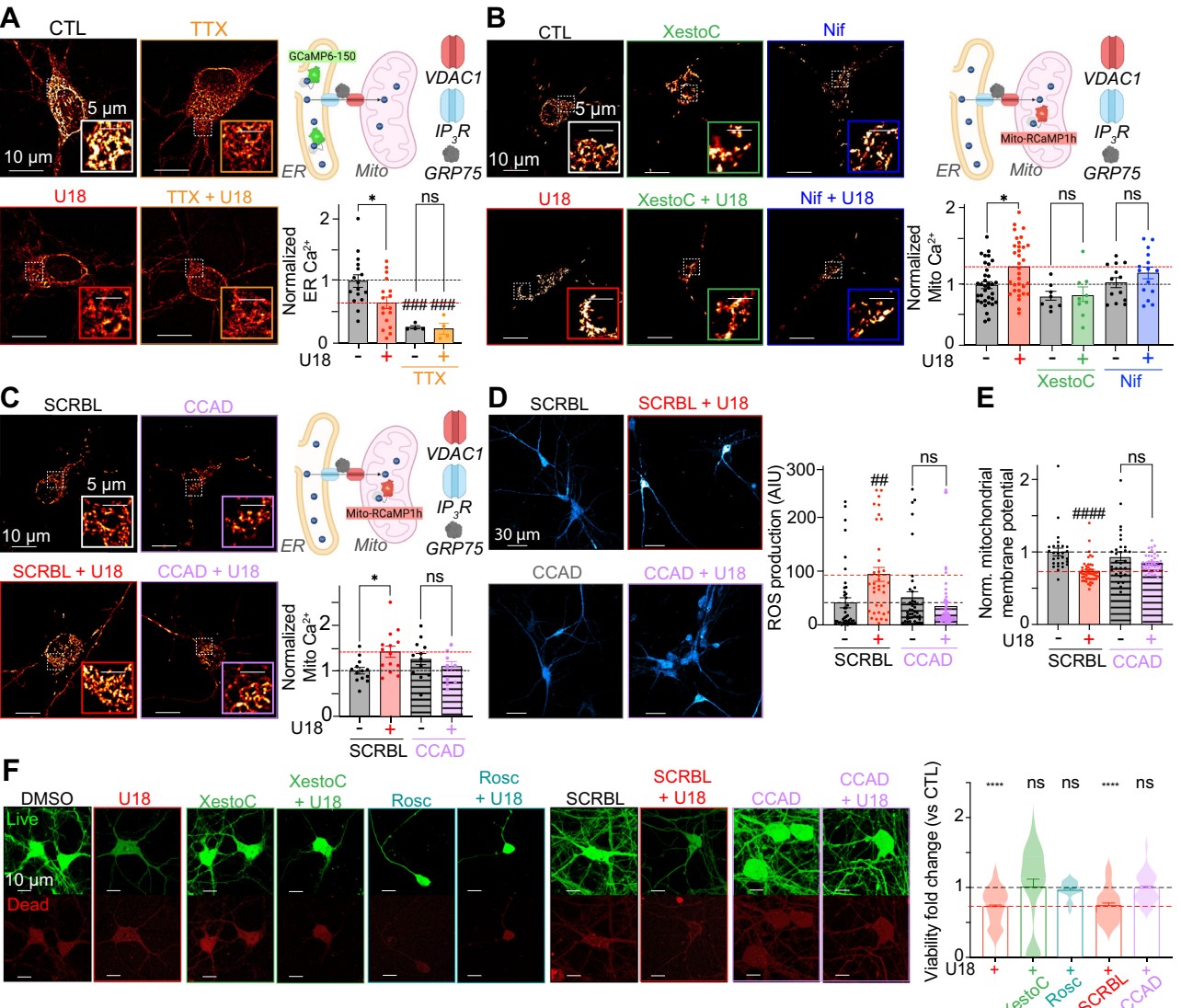

**Fig. 10 | Targeting $K_V2.1$–$Ca_V1.2$ interactions reduces Mito $Ca^{2+}$ and rescues neurotoxicity in NPC disease. A** *Left*, representative images of neurons transfected with ER-GCaMP6-150. *Top right*, ER-GCaMP6-150 localization. *Bottom right*, ER $Ca^{2+}$ quantifications. $N = 16$ (CTL), $n = 17$ (U18), $n = 4$ (TTX) and $n = 4$ (TTX + U18) neurons were analyzed across 2 independent isolations. **B** *Left*, representative images of CTL (black), U18 (red), nifedipine (Nif) (blue) and Xestospongin C (XestoC) (green) neurons transfected with Mito-RCaMP1h. *Right top*, Mito-RCaMP1h localization. *Right bottom*, normalized Mito $Ca^{2+}$. $N = 35$ (CTL), $n = 33$ (U18), $n = 13$ (Nif), $n = 15$ (Nif+U18), $n = 8$ (XestoC) and $n = 9$ (XestoC+U18) neurons were analyzed across 3 independent isolations. **C** *Left*, neurons expressing Mito-RCaMP1h. *Right top*, Mito-RCaMP1h localization. *Right bottom*, normalized Mito $Ca^{2+}$. $N = 13$ (SCRBL), $n = 14$ (SCRBL + U18), $n = 12$ (CCAD) and $n = 12$ (CCAD + U18) neurons were analyzed across 2 independent isolations. **D** *Left*, representative images of neurons incubated with H2DCFDA. *Right*, quantification of ROS. $N = 43$ (SCRBL), $n = 36$ (SCRBL + U18), $n = 39$ (CCAD) and $n = 64$ (CCAD + U18) neurons were analyzed

across 2 independent isolations. **E** Quantification of neuron mitochondrial membrane polarization. $N = 28$ (SCRBL; black), $n = 48$ (SCRBL + U18; red), $n = 30$ (CCAD; black) and $n = 39$ (CCAD + U18; purple) neurons were analyzed across 2 independent isolations. **F** Left, representative images of Live and dead channels from neurons in each condition. Quantification of neuronal viability. $N = 394$ (DMSO), 174 (U18), $n = 94$ (XestoC), $n = 28$ (XestoC + U18), $n = 124$ (SCRBL), $n = 50$ (SCRBL + U18), $n = 85$ (CCAD), $n = 59$ (CCAD + U18), $n = 33$ (Rosc) and $n = 29$ (Rosc + U18) neurons were analyzed across 5 independent isolations. All error bars represent SEM. Statistical significance was calculated using a two-way ANOVA test in (**A**)–(**E**) and One sample $t$ test and Wilcoxon test in (**F**). ns: not significant; *$P < 0.05$; **$P < 0.01$; ***$P < 0.001$; ****$P < 0.0001$. ##$P < 0.01$; # indicates comparison with the CTL condition. SCRBL is scramble, U18 is U18666A, CCAD is calcium channel association domain, Rosc is Roscovitine, TTX is tetrodotoxin, Nif is Nifedipine, XestoC is Xestospongin C.

In addition to physically interacting to organize $Ca_V1$ channels, $K_V2$ also functionally coordinates RyR and SERCA at ER–PM nanodomains[2,31,35]. We find that loss of NPC1 function positively remodels the distribution of these proteins at ER–PM MCSs to increase $Ca^{2+}$ entry into the cytoplasm. It is noteworthy that as $K_V2.1$ cluster size and density increase so does the number of ER–PM MCSs. This observation aligns with reports that phosphorylated $K_V2.1$ interacts with the ER VAP proteins to generate ER–PM MCSs[34,35]. However, our data suggest that it is not simply the physical coupling

of $K_V2.1$–VAP that defines this sub-population of ER–PM MCSs, as uncoupling $Ca_V1$ from $K_V2$ channels with CCAD, a perturbation that maintains $K_V2.1$–VAP interactions[2], decreases the overall number and density of ER–PM MCSs. That uncoupling of $K_V2$–$Ca_V1$ channels reduces the amount of $Ca^{2+}$ entry into the cytoplasm (Fig. 8C and ref. 2) supports that $K_V2$–$Ca_V1$ dependent $Ca^{2+}$ entry is either a stimulus for stabilizing ER–PM MCS or an instructive signal to induce the formation of new ER–PM MCSs in neurons. The latter model that $Kv2$–$Ca_V1$ $Ca^{2+}$ entry nucleates the formation of new MCSs aligns with

previous reports that a number of lipid binding ER–PM MCS forming proteins are sensitive to local changes in cytoplasmic $Ca^{2+}$ concentrations[29,94] and provides evidence that $K_V2$ channels are a crucial foundational element that regulates multiple sub-populations of ER–PM MCSs.

In the present study, we find that NPC1 deficiency leads to increased proximity between $Ca_V1.2$–RyR and $K_V2.1$–SERCA, suggesting that $K_V2.1$-associated ER–PM MCSs are enriched in RyR, SERCA and $Ca_V1.2$, consistent with previous proteomics analyses of Kv2.1-containing ER–PM MCSs purified from mouse brain[31]. Having more $Ca_V1.2$ juxtaposed to RyR would amplify $Ca^{2+}$ entry into the cytoplasm through $Ca^{2+}$-Induced $Ca^{2+}$-Release (CICR). Such amplification of $Ca^{2+}$ signals is important in normal neuronal function as it facilitates excitation-transcription coupling[2], however, in NPC disease it is further increased and together with the hyperexcitability phenotype results in aberrant and ultimately neurotoxic (see below) increases in $Ca^{2+}$. An intuitive prediction of enhanced $K_V2$–$Ca_V1$–RyR $Ca^{2+}$ entry in NPC disease would be an increase in ER $Ca^{2+}$ levels. The statement is supported by the increase in clustering at ER–PM MCSs of two ER $Ca^{2+}$ uptake mediator proteins, $K_V2.1$ and SERCA (Fig. 3 and S7A)[75], and increased SERCA activity in NPC disease models[23]. Thus, it was initially surprising that despite the alignment of these factors, ER $Ca^{2+}$ levels are significantly decreased in NPC disease (Fig. 10A and ref. 23). The molecular mechanism underlying reduced ER $Ca^{2+}$ in NPC disease is enhanced spontaneous $Ca^{2+}$ release from $IP_3R1$ in ER membranes[17]. Evidence herein suggests that heightened RyR CICR activity may also contribute to the leaky ER $Ca^{2+}$ phenotype in NPC disease.

A common feature in neurodegenerative disorders is mitochondrial $Ca^{2+}$ overload leading to mitochondrial membrane rupture and release of pro-apoptotic factors[17,77,95–98]. The $IP_3R$–GRP75–VDAC1 signaling axis is a hotspot for $Ca^{2+}$ flux from the ER towards the mitochondria. Our data provide evidence that ER $Ca^{2+}$ stores are low whereas mitochondrial $Ca^{2+}$ is significantly increased in NPC neurons. Furthermore, we find that increases in mitochondrial $Ca^{2+}$ correlate with alterations in MMP, increased ROS generation, and cell death supporting the model that mitochondrial $Ca^{2+}$ overload is a mechanism for neurodegeneration in NPC disease. Mechanistically, increased interactions between $IP_3R$–GRP75–VDAC1 at ER–Mito MCSs coupled with more spontaneously active $IP_3R1$[17] creates the opportunity for destructive funneling of $Ca^{2+}$ into mitochondria. We propose that in NPC disease, excess $K_V2$–$Ca_V1$ $Ca^{2+}$ entry at ER–PM MCSs is transported into the ER through increased SERCA activity[23] and clustering but instead of accumulating to increase ER $Ca^{2+}$ concentrations, is quickly released by $IP_3R1$ receptors at ER–Mito MCSs to promote neurotoxic increases in mitochondrial $Ca^{2+}$ (Fig. S10). Supporting this model, we demonstrate that disrupting $K_V2$–$Ca_V1$ interaction or reducing CDK5 phosphorylation of $K_V2$ channels reduces mitochondrial $Ca^{2+}$ levels and rescues neuronal death following NPC1 inhibition, supporting the concept that $K_V2$–$Ca_V1$ $Ca^{2+}$ entry at ER–PM MCSs is an upstream contributing factor to neurotoxic increases in mitochondrial $Ca^{2+}$ in NPC. Future investigations should lead to a better understanding of whether disrupting $Ca^{2+}$ entry at $K_V2$–$Ca_V1$ nanodomains has beneficial effects on disease progression in NPC disease models, as it does for stroke models[66,99]. Noteworthy, previous reports have linked altered $K_V2.1$ activity to cell death through increased $K^+$ efflux activating pro-apoptotic pathways[100,101] rather than by favoring $Ca^{2+}$ entry as we describe for NPC disease. Together, these data reinforce the need for homeostatic control of conducting ($K^+$ efflux) and non-conducting ($Ca^{2+}$ nanodomain organization) $K_V2.1$ subpopulations by protein kinases and phosphatases to ensure the maintenance of neuronal health.

Why is there such a prominent $Ca^{2+}$ phenotype across multiple organelles in NPC disease? Our working model is that alterations in $Ca^{2+}$ gradients across MCSs represent cellular programs attempting to redistribute cholesterol rapidly and efficiently to cellular membranes to restore cholesterol homeostasis[93,102,103]. In this model, unless new homeostatic set points are found quickly, gross alterations in $Ca^{2+}$ gradients, as well as lipids[16,52], provide a substrate for neurodegeneration[17,23]. To design rational targets to slow or abrogate NPC-linked pathology, further quantitative characterization of the structural and functional alterations to ER–PM contacts and the downstream consequences linking mitochondrial $Ca^{2+}$ elevations to neurodegeneration are required.

To conclude, this study details a molecular link between cholesterol egress from lysosomes and the reorganization of $Ca^{2+}$ handling proteins at two membrane contact sites: ER–PM and ER–Mito. Our data demonstrate that NPC is a nanostructural ion channel clustering disease with altered ion channel distribution/activity at membrane contacts contributing to neurodegeneration.

## Methods

### Key resources
Key resources are documented in Supplementary Table 1.

### Animals
All experiments were performed in strict compliance with the University of California Davis ethical regulations for studies involving animals as approved by the University of California Davis Animal Care and Use Committee (protocol #: 22644). C57BL/6 WT and NPC1$^{II061T}$ mice were purchased from The Jackson Laboratory and kindly provided by Daniel Ory[64], respectively. Mice were maintained under standard light-dark cycles, fed standard chow and water ad libitum, and housed in a vivarium with controlled conditions.

### Cell culture
tsA201 cells were purchased from Sigma (Cat #96121229), CHO WT and NPC1$^{-/-}$ cells were a kind gift from Dr. Ory (Washington University, St. Louis, MO), HeLa WT and NPC1$^{-/-}$ cells were a kind gift from Dr. Judith Storch (Rutgers). tsA201 and HeLa cells were cultured in DMEM supplemented with 10% FBS (GIBCO, Cat #26140-079), with 2 mM L-glutamine (GIBCO, Cat #25030-081) and 0.2% Penicillin/Streptomycin (GIBCO, Cat #15140-122). CHO cells were cultured in DMEM/F12 (1:1) (GIBCO, Cat #11320033) supplemented with 10% FBS (GIBCO, Cat #26140-079) and 0.2% Penicillin/Streptomycin (GIBCO, Cat #15140-122). All cell lines were passaged twice weekly and incubated in 5% $CO_2$ at 37 °C. NPC1$^{-/-}$

Cortical neurons from mice of both sexes were dissociated at embryonic day 15–18 (E15–18) using the Papain Dissociation System purchased from Worthington (Cat #LK003150). All stock solutions were prepared as per manufacturer's recommendations. Dissection was conducted in sterile PBS at 4 °C as previously described in[104] and meninges, cerebellum, hippocampus, and striatum were discarded. Cortical tissue was pelleted and incubated in papain for 20 min at 37 °C (agitating every 5 mins), followed by trituration. Consequently, cells were centrifuged 5′ at 1000$g$ and resuspended in Earle's Balanced Salt Solution (EBSS), ovomucoid (papain inhibitor) and DNase. Neurons were plated in Neurobasal (21103-049; Gibco) supplemented with B27 (GIBCO, Cat #17504-044), Glutamax (GIBCO, Cat #35050-061) and 0.2% penicillin/streptomycin at a density of 650,000 cells / mL. Neurons were incubated in 5% $CO_2$ at 37 °C and half of the media was changed every 4 days. Experiments were performed at DIV 6-8, except for the GCaMP3-$K_V2.1_{P404W}$ $Ca^{2+}$ imaging which was conducted at DIV 14–16. Rat cortical neurons were dissected and isolated at gestation day 18 as previously described in Vierra et al.[31].

### Reagents
Xestospongin C (XestoC) was dissolved in DMSO and used at a final concentration of 10 μM and incubated overnight (O/N) or for 48 h. Torin-1 was dissolved in DMSO and used at a final concentration of 10 μM and incubated O/N. U18666A (U18) was dissolved in DMSO to a final concentration of 10 μM and incubated O/N or for 48 h.

Tetrodotoxin (TTX) was dissolved in DMSO and incubated O/N at a final concentration of 200 nM. roscovitine (Rosc) was dissolved in DMSO and incubated O/N or for 48 h at a final concentration 10 μM. Ionomycin was dissolved in DMSO and used at a final concentration of 2.5 μM for ~6 min. TAT-FFAT-HA (472-481) (FFAT), TAT-HA-C1aB (CCAD) and their respective control scramble peptides TAT-HA-C1aB-Scr and TAT-FFAT-HA (472-481)-Scr (SCBRL) were dissolved in molecular biology grade water and used at a final concentration of 1 μM for 48 h.

## Fillipin staining

Cells were washed with PBS and fixed in 4% paraformaldehyde (Electron Microscopy Sciences, Cat #15710) for 20 min at 21 °C. Cells were stained with fillipin (100 μg/ml in PBS) for 2 h at room temperature (RT, 21–23 °C). During the fillipin staining process cells were protected from light exposure. Cells were excited using a 405-nm LED and light collected using a Plan-Apochromat 63×/1.40 oil-immersion lens and a Zeiss LSM 880 Airyscan microscope. Images were acquired in PBS solution at RT using Zen v2.3 software.

## Protein extraction and abundance determination

Protein was harvested from neuronal cultures in RIPA buffer (Thermo Scientific, Cat #89900) containing Complete/Mini/EDTA-free protease inhibitor cocktail (Roche, Cat #11836170001), sodium fluoride 1 mM (Sigma-Aldrich, Cat #67414), SDS 0.05%, sodium deoxycholate 0.4% (DOC) and Microcystin 4 μg/mL (Sigma Millipore, Cat #475821). Cells were scraped in 100 μL of lysis buffer and centrifuged at 13,200 rpm for 20 min at 4 °C to isolate the postnuclear supernatant. Protein concentration was quantified using a Pierce BCA protein assay kit (Thermo Scientific, Cat #23225). 15 μg of each protein sample was added to 4%–12% Bis-Tris gels and electrophoresed under reducing conditions for approximately 75 min at 155 V. Proteins were transferred onto nitrocellulose membranes (Life Technologies, Cat #LC2000) using the Mini-Blot system (Thermo Scientific, Cat #A25977) for 2 h 30 min at 12 V (~200 mA). After 30 min incubation at RT in Tris-Buffered Saline (TBS) buffer supplemented with 0.05% Tween-20 (TBS-T) and 5% non-fat dry milk, membranes were exposed O/N at 4 °C to the following primary antibodies: $K_V2.1$ at 4 μg/mL (NeuroMab, K89/34), $Ca_V1.2$ at 1:250 (Alomone Labs, acc-003), CDK5 at 1:100 (Santa Cruz Biotechnology, Cat #sc-6247), p35 at 1:100 (Cell Signaling Technology, Cat #C64B10), p39 at 1:100 (Santa Cruz Biotechnology, Cat #sc-365781) or GAPDH at 1:1000 (Proteintech, Cat #10494-I-AP). Blot bands were detected using fluorescent secondary antibodies goat anti-rabbit 680RD at 1:1000 (LI-COR, Cat #P/N 926-68071) and goat anti-Mouse 800CW at 1:1000 (LI-COR, Cat #P/N 925-32210). Signals were detected using Azure Sapphire™ Biomolecular Imager (Azure Biosystems) and quantified using the ImageJ software. Abundance of proteins was normalized to GAPDH.

## Transfection and plasmids

Plasmid transfection mixtures were prepared using 2.5 μl/ μg DNA of Lipofectamine 2000 (Invitrogen, Cat #11668-019), added to neurons and incubated for 2 days. The following plasmids (per 650,000 neurons) were used in this study: 4 μg pCAG-mito-RCaMP1h (Addgene, Cat #105013), 4 μg ER-GCaMP6-150 (Addgene, Cat #86918), 2 ug ArcLight-Q239 (Addgene, Cat #36856)[76], 2 μg GFP-MAPPER[74], 1 μg SPLICS$_L$-P2A$^{ER-PM}$ (Addgene, Cat #164111), 1 μg SPLICS$_S$-P2A$^{ER-PM}$ (Addgene, Cat #164112) and 1.1 μg GCaMP3-Kv2.1$_{P4O4w}$[31]. Plasmid transfection mixtures for CHO cells were prepared using 7 μl of Lipofectamine LTX and 7 μl of Plus (Invitrogen, Cat #15338-030) and were imaged the day after transfection. The following plasmids were used: 1 μg per 960,000 cells rat $Ca_V1.2α1c$ (GenBank number NP_001129994.1; a gift from Dr. William Catterall, University of Washington, Seattle, WA), 1 μg per 960,000 cells rat $Ca_Vα_2δ$ (AF286488; a gift from Dr. Diane Lipscombe, Brown University,

Providence, RI), 1 μg per 960,000 cells $Ca_Vβ_3$ (M88751; a gift from Dr. Diane Lipscombe, Brown University, Providence, RI) and 1 μg per 960,000 cells DsRed-$K_V2.1$[35].

## Super resolution airyscan imaging

Imaging was conducted at RT using a Zeiss LSM880 confocal laser scanning microscope equipped with a Plan-Apochromat 63× 1.4 Oil DIC M27 objective and a super-resolution Airyscan detection unit. Images of GFP/Alexa-488, Alexa-568/mCherry and Alexa-647 were collected using 488-, 594- and 633-nm excitation lasers, respectively, with multicolor images sequentially acquired. Fixed and live cells were acquired in PBS or Ringer's solution (in mM, 160 NaCl, 2.5 KCl, 2 $CaCl_2$, 1 $MgCl_2$, 10 HEPES, and 8 D-Glucose) with the ZEN software v2.3, respectively. Images were taken at 0.5 μm depth intervals within the cell (see PLA assay and ER staining in live neurons) or as a single internal or plasma membrane plane for HA, GRP75, IP$_3$R and VDAC1 or, $Ca_V1.2$, $Ca_V1.3$, $K_V2.1$, $Ca_V2.1$ and RyR immunolabeling, respectively. 3D analysis for the ER morphology in neurons was conducted with the IMARIS software (Oxford Instruments); all other datasets were carried out following background subtraction and similar thresholding using ImageJ (NIH, Bethesda, MD, USA). Colocalization/overlap analyses were performed by multiplying binary masks of $Ca_V1.2$, $K_V2.1$, RyR, IP$_3$R, VDAC1, ER and mitochondria images. For neuronal analyses, the soma area and a 15 μm$^2$ ROI from proximal neuronal dendrites were used. For protein clustering and ER–PM MAPPER puncta studies in the soma and cell body, sum intensity and numbers of clusters were normalized to their respective areas.

## Super resolution TIRF imaging

Images were acquired using a Leica Infinity TIRF microscope equipped with a 163× 1.49 (fixed cells) and a 100× 1.47 (live cells) CORR TIRF oil immersion objective and a Hamamatsu orca flash 4.0 camera. Leica LAS X software was employed for both image acquisition and processing. GFP/YFP, Alexa-647 and Alexa-568 images were collected with the 488 nm, 638 nm and 561 nm excitation lines, respectively. Fixed or live cells were imaged in a GLOX–MEA (Tris 10 mM pH8, Glucose Oxidase 56 mg/mL, Catalase 34 mg/mL, 10 mM MEA) or Ringer's solution (in mM, 160 NaCl, 2.5 KCl, 2 $CaCl_2$, 1 $MgCl_2$, 10 HEPES, and 8 D-Glucose), respectively. For fixed cell imaging ($K_V2.1$–VAPA/B, $K_V2.1$–CDK5, $Ca_V1.2$, $K_V2.1$, $K_V2.1$–$Ca_V1.2$ and $K_V2.1$–SERCA) 50,000 cycles with an exposure time of 10 ms were acquired and imaging analysis was performed using IMARIS software (Oxford Instruments). For all analyses, cluster number was normalized to the respective soma area. Colocalization/overlapping analyses were performed by creating a 2D $K_V2.1$ mask and measuring $Ca_V1.2$, CDK5, VAPA/B or SERCA areas within said mask. Live cell imaging of GCaMP3-$K_V2.1_{P4O4W}$ analyses were performed using the ImageJ and ClampFit (Molecular Devices) softwares.

## Immunofluorescence immunocytochemistry

Antibody dilutions/amounts, validation, company names, catalogue numbers and clone numbers can be found either below, in the reporting summary, and/or supplementary table 1. Cells were fixed for 20 min at RT in PBS with 4% formaldehyde freshly prepared from paraformaldehyde and blocked for 1 h in 50% SEA BLOCK and 0.5% Triton X-100 solution. Neurons were incubated O/N at 4 °C in 20% SEA BLOCK and 0.5% Triton X-100 in PBS with the following primary antibodies: GRP75 1:100 (Abcam, Cat #ab2799), $K_V2.1$ 10 μg/mL (NeuroMab, K89/34), $Ca_V1.2$ 1:333 (Alomone, Cat #acc-003), $K_V2.1$(pS603) 1:5 (L61/14.2), VDAC1 1:100 (Abcam, Cat #ab14734), SERCA 10 μg/mL (Abcam, Cat #ab2861), $Ca_V1.3$ 10 μg/mL (Alomone Labs, Cat #ACC-005), pan-RyR 1:100 (Abcam, Cat #ab2868), IP$_3$R 18 μg/mL (Abcam, Cat #ab5804), $Ca_V2.1$ 1:200 (Alomone Labs, Cat #ACC-001), and $K_V2.1$ pS603 1:5 (L61/14). After primary antibody incubation, neurons were washed 3 × 5 min and subsequently incubated for 1 h at RT with the

following secondary antibodies: Goat anti-Mouse-647 and −568 nm 1:1000 (Invitrogen, Cat #A21236 and Cat #A11031, respectively), anti-Rabbit-647 and −555 nm 1:1000 (Invitrogen, Cat #A21245 and Cat #A21429, respectively), anti-Mouse IgG1-568 nm 1:1000 (Invitrogen, Cat #A21124) and anti-Mouse IgG1-CF568 1:250 (Sigma-Aldrich, Cat #SAB4600314).

## Immunofluorescent labeling of brain sections

*Fixation:* Postnatal day 60 (P60) wild-type and NPC1[II016T] mice were anesthetized and transcardially perfused with 4% phosphate buffered (PB) formaldehyde (pH 7.4) made from powdered paraformaldehyde using a peristaltic pump. Perfused brains were collected and cryo-protected with 30% sucrose/PB solution. *Sectioning*: Sagittal brains sections (30 μM) were generated using a freezing microtome and immunolabeling was performed on free floating sections. Briefly, tissue was blocked in 10% normal goat serum (NGS) with 0.3% TritonX-100 in 0.1 M PB at RT for 1 h followed by incubation in primary antibody O/N at 4 °C. Primary antibody incubation was performed in 5% NGS/0.3% TritinX-100/0.1 M PB. *Antibodies:* the following primary antibodies were used for immunolabeling: Calbindin 1:200 (Sigma-Aldrich #C9848),Calbindin, 2 μg/mL (L109/57, NeuroMab), $K_V$2.1 10 μg/mL (NeuroMab, K89/34), $Ca_V$1.2 1:200 (Alomone Labs, Cat #acc-003), $Ca_V$1.2 5 μg/mL (NeuroMab, N263/31 R). Species-specific Alexa Fluor 488, 555 and 647conjugated secondary antibodies (2 μg/mL; Life Technologies) were incubated for 60 min at RT. Images were taken using a Zeiss ApoTome Fluoview FV3000 confocal microscope (Olympus) or a Zeiss Axioscope 2 widefield microscope with Apo-Tome. All images were assembled using FIJI and Illustrator (Adobe). $Ca_V$1.2–$K_V$2.1 colocalization volumes were calculated using IMARIS software by quantifying the $K_V$2.1-positive pixels inside a 3D $Ca_V$1.2 mask.

## PLA assay

The Duolink In Situ PLA kit (Sigma-Aldrich, Cat #DUO92004-100RXN and DUO92002-100RXN) was used to quantify $Ca_V$1.2-$K_V$2.1 proximity in fixed cortical neurons treated with U18 or vehicle control (DMSO). Following fixation, as described in *Immunofluorescence Immunocytochemistry* above, cells were incubated for 15 min with 100 mM glycine at RT. Subsequently, neurons were washed in PBS (2 ×3 min) and permeabilized in 0.1% Triton-X100 for 20' at RT. Neurons were blocked in a 20% SEA BLOCK solution (Thermo Scientific, Cat #37527) for 1 h and incubated at 4° C O/N with the following primary antibodies: $K_V$2.1 10 μg/mL (NeuroMab, K89/34) and $Ca_V$1.2 1:333 (Alomone, Cat #acc-003;1:333). Secondary oligonucleotide-conjugated antibodies (PLA probes: anti-mouse MINUS and anti-rabbit PLUS) were used at 1:5 dilution in Duolink Antibody Diluent. PLA assay was performed as per the manufacturer's recommendations. Coverslips were mounted with DAPI Fluoromount-G (SouthernBiotech, Cat #0100-20). The fluorescence signal was detected using a Zeiss LSM880 confocal laser scanning microscope equipped with a Plan-Apochromat 63× 1.4 Oil DIC M27 objective and a super-resolution Airyscan detection unit. 405- and 488-nm lasers were employed to visualize the DAPI-stained nucleus and PLA signal, respectively. For each neuron, optical *z*-axis sections were acquired at 0.5 μm intervals and combined to a single maximum intensity projection using ImageJ.

## ER and mitochondrial staining in live neurons and tsA201 cells

Cells were incubated for 20 min using Mito Tracker (50 nM; Invitrogen, Cat #M22426) and ER Tracker (100 nM; Invitrogen, Cat #E12353). Cells were subsequently washed twice (5 min each time) before being imaged with a Zeiss LSM880 confocal laser scanning microscope equipped with a Plan-Apochromat 63× 1.4 Oil DIC M27 objective and a super-resolution Airyscan detection unit; 633- and 546-nm excitation lasers were used to image mitochondria and ER, respectively. For tsA201, cells single planes near the PM were acquired and overlapping

areas between both organelles were analyzed by multiplying their respective binary masks in ImageJ and normalized to the respective cell body area. For neurons, z-stacks were taken at 0.5 μm depth intervals within the cell and analysis was performed in the IMARIS software (Oxford instruments).

## Mitochondrial and ER $Ca^{2+}$ imaging

48 h post-transfection with pCAG-mito-RCamPh1 or ER-GCaMP6-150, neurons were imaged in a 2 mM $Ca^{2+}$ Ringer's solution using a LSM880 confocal laser scanning microscope equipped with a Plan-Apochromat 63× 1.4 Oil DIC M27 objective and a super-resolution Airyscan detection unit. A 546 nm laser was used to excite pCAG-mito-RCamPh1, while a 488 nm laser was used to excite ER-GCaMP6-150. After collection of a baseline image, cells were perfused with a 20 mM Ringer's solution containing 2.5 μM Ionomycin for 6 min. The mean intensity ratio of pre-ionomycin/post-ionomycin signals was calculated for each cell. Higher ratios were taken to indicate higher resting mitochondrial or ER $Ca^{2+}$ concentrations.

## Cytosolic $Ca^{2+}$ imaging

Cells were incubated in a 2 mM $Ca^{2+}$ Ringer's solution containing 2.5 μM Fluo-4 and 0.1% Pluronic acid for 30 min. Cells were then moved to a Fluo-4−free Ringer's solution to deesterify the Fluo-4 for 30 min. Following deesterification, Fluo-4−loaded cells were excited with a 488-nm laser and the resulting fluorescence monitored using an inverted microscope with a Plan-Apochromat 63×/1.40 oil objective, connected to an Andor W1 spinning-disk confocal with a Photometrics Prime 95B camera. Images were acquired every 50 ms for a total of 50 s in a 2 mM $Ca^{2+}$ Ringer's solution at RT using Micromanager (1.4.21) software. Electrical stimulation at 40 V was performed at 1 Hz for 5 s with a Field Stimulator (IonOptix). Intracellular $Ca^{2+}$ activity was measured as $Ca^{2+}$ peak frequency and $Ca^{2+}$ peak amplitude in a Region of Interest (ROI) localized in the soma or proximal neuronal dendrites using ImageJ or ClamPfit software.

## GCaMP3-$K_v$2.1$_{P4O4W}$ imaging

48 h post-transfection, TIRF imaging was performed using a Leica Infinity TIRF super-resolution microscope equipped with a 100× 1.47 TIRF oil immersion objective and a Hamamatsu Orca Flash 4.0 camera. Images were acquired in Ringer's solution (in mM, 160 NaCl, 2.5 KCl, 2 $CaCl_2$, 1 $MgCl_2$, 10 HEPES, and 8 D-Glucose) containing B-Kay 500 nM every 100 ms for a total time of 50–100 s by exciting with a 488-nm laser. $Ca^{2+}$ activity was measured as *GCaMP3-$K_v$2.1$_{P4O4W}$* peak amplitude in a *ROI* localized in the soma or proximal neuronal projections using ImageJ and ClamPfit softwares.

## H2DCFDA assay (ROS production)

The ROS production assay was conducted as per the manufacturer's recommendations (Invitrogen, Cat #D399). Live DIV 6-8 cortical neurons were incubated O/N with U18 and CCAD or SCRBL peptides. Single-plane images were collected in Ringer's solution (in mM, 160 NaCl, 2.5 KCl, 2 $CaCl_2$, 1 $MgCl_2$, 10 HEPES, and 8 D-Glucose) at 488 nm using a Plan-Apochromat 63× 1.4 Oil DIC M27 objective and a Zeiss 880 Airyscan microscope at RT. Images were analyzed using ImageJ and mean intensities were taken to plot ROS production.

## *MitoProbe JC-1 assay* (mitochondrial membrane potential)

Mitochondrial membrane potential was conducted as per the manufacturer's recommendations (Abcam, Cat #ab113850) and plotted as a ratio of aggregated JC-1 form/monomer JC-1 form, with higher ratios indicating more polarized membranes. JC-1 aggregates were imaged at a 618 nm emission and a 488 nm excitation, and JC-1 monomers at a 536 nm emission and 488 nm excitation in Ringer's solution (in mM, 160 NaCl, 2.5 KCl, 2 $CaCl_2$, 1 $MgCl_2$, 10 HEPES, and 8 D-Glucose) using a Plan-Apochromat 63× 1.4 Oil DIC M27 objective

and a Zeiss 880 Airyscan microscope at RT. Images were analyzed using ImageJ software.

## Cell viability assay

The cell viability assay was conducted as per the manufacturer's recommendations (K502-100; BioVision). Live DIV 6–8 cortical neurons were incubated 48 h with U18, XestoC, Rosc or CCAD or SCRBL peptides before being washed once with 2 mM $Ca^{2+}$ Ringer's solution and loaded with 1 mL assay buffer containing 2 μL Live cell staining dye and 1 μL Dead cell staining dye. Cells were immediately excited using a 488- and 564- nm LED. Single-plane images were collected using a Plan-Apochromat 63× 1.4 Oil DIC M27 objective and a Zeiss 880 Airyscan microscope at RT. Images were analyzed using ImageJ. The ratio of live (488 nm) to dead (564 nm) was taken for each single neuron every captured image, with higher ratios indicating increased neuronal viability.

## Statistics and reproducibility

The number of replicates is based on previously published observations to reached statistical differences between datasets[16,17,23,26,52,104–107]. When comparing two or three independent groups, normality was determined with the Agostino-Pearson omnibus test and a parametric, unpaired $t$ test, or nonparametric, Mann–Whitney $t$ test, was employed. When comparing the means of 4 groups with two or more categorical independent variables, the two-way ANOVA for multiple comparisons with appropriate post hoc test was used. When quantifying fold changes respective to the control condition, one sample $t$ test was employed. $P < 0.05$ was considered statistically significant (*$P \leq 0.05$, **$P \leq 0.01$, ***$P \leq 0.001$, ****$P \leq 0.001$); # shows significance between sample groups and the negative control when ANOVA was employed. To determine whether values within a dataset have significant outliers, a two-sided Grubbs' test, with an alpha value of 0.05 was performed. All the data values are presented as means ± SEM and the number of technical and biological replicates is detailed in each figure legend, which is based on previously published observations and is consistent with the general number of replicates and conditions commonly accepted in the field. For each neuronal dataset experiments were performed from 1-4 independent neuronal cultures with each isolation containing 6–8 pups. Several key datasets were blinded and outcome accessed.

## Reporting summary

Further information on research design is available in the Nature Portfolio Reporting Summary linked to this article.

# Data availability

The data supporting the findings of this work can be found in the source data file and additional raw data will be made available upon request. Source data are provided with this paper.

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

## Acknowledgements

The authors are extremely grateful to those laboratories that shared reagents, plasmids, and cells lines used in this study. This work was supported by an Ara Parseghian Medical Research Foundation award (E.J.D.), NIH grants R01 GM127513 and R35 GM149211 (E.J.D.), NIH grant R01 NS131375 (E.J.D and S.S.), NIH grant F32 NS108519 (N.C.V), NIH grant R01 NS109176 (S.S.), NIH grants R01 AG063796 and R01 HL159304 (R.E.D.), and NIH grant R01 NS114210 (J. S. T.). Illustrations were generated using Biorender.com.

## Author contributions

M.C., K.D.M., K.H., and E.J.D. performed experiments. M.C. and E.J.D. performed image analysis. M.C., K.D.M., N.C.V., S.S., J.S.T., R.E.D., and E.J.D. designed experiments. All authors contributed to the writing of the manuscript.

## Competing interests

The authors declare no competing interests.
