## [Peer Review File · Nature Communications]

NPC1-dependent alterations in KV2.1–CaV1 nanodomains drive neuronal deathREVIEWER COMMENTS

Reviewer #1 (Remarks to the Author):

This manuscript uncovers a new Ca²⁺ phenotype resulting from NPC1 deficiency. The study elegantly demonstrates clustering of Kv2.1 and CaV1.2 channels at the PM in the absence of functional NPC1, which promotes Ca²⁺ influx to mitochondria via the ER. The findings are novel, timely and of interest to a wide audience. The manuscript is very clearly written, with excellent data presentation (I particularly like the illustrations behind each figure). The findings advance our understanding both of calcium signalling and NPC pathophysiology, but I do have a few comments and concerns:

1. The hypothesis is clear and the data support the conclusions but there is a lack of mechanistic detail linking loss of NPC1 function to events at the PM. The authors implicate CDK5 in serine phosphorylation of KV2.1 to increase affiliation with CaV1.2 and promote clustering at the PM which in turn influences PM contact with the ER. How NPC1 function regulates CDK5 activity or how clustering of the channels regulates MCS is not adequately investigated.

- Is CDK5 upregulated in NPC models (a WB would be nice!)? If so, how? Since mTORC1 is hyperactivated by high cholesterol in NPC, this is a likely candidate (ie, activation of mTORC1 leading to inhibition of AMPK and upregulation of CDK5). Does inhibition of mTORC1 (eg Torin1) have the same effect on U18-mediated clustering as CDK5 inhibition?
- Does S603 phosphorylation affect VAP binding or is this effect on MCS independent of KV2.1-VAP interaction? S603 is approx 10a/a downstream of KV2.1's noncanonical FFAT motif. Is KV2.1-VAP interaction required for the effects on Ca²⁺ signalling - do you still see Ca²⁺ influx in cells depleted of VAPs?

2. Do you see evidence of mitochondrial dysfunction(eg. mtROS, membrane potential etc) in your models of NPC and if so can you relate it to the channel clustering - ie, does TAT-CCAD rescue mitochondrial dysfunction?

3. Thapsigargin is used to inhibit SERCA-mediated Ca²⁺ sequestration into the ER, preventing elevations in mitochondrial Ca²⁺ which are proposed to result from the increased channel clustering at the PM. However, since SERCA inhibition also results in increased cytosolic [Ca²⁺], wouldn't thapsigargin be predicted to promote mitochondrial calcium uptake? If including the thapsigargin data, this needs a bit of explanation - could thapsigargin be activating calcineurin-mediated Kv2.1 dephosphorylation to downregulate clustering etc? It might be over-complicating things but could fairly easily be checked with anti-pS603?

4. Does inhibition of clustering-mediated calcium influx (eg inhibition of CaV1 or IP3R) prevent higher frequency APs? I don't know how involved these experiments are - would make a nice addition but the story is also strong without!

5. Minor points:

- There is possibly an over-reliance on NPC1 inhibition by U18666A, which has been reported to have off-target effects. However, validation both of some data and the use of U18 by results obtained from other models of NPC1 deficiency provides a high level of confidence in the findings.
- p13, line 351 says that results demonstrate a role for NPC1 in organizing coupling between channels etc. This is slightly misleading as, unless I have misunderstood, NPC1 is not directly involved in organizing the channels. Maybe it could be softened to demonstrating novel downstream effects of NPC1 function(NPC1-mediated lipid transport?) in regulating the organization of channels at the PM?
- p13, line 419 Thapsigargin typo.
- p18, line 494 questions why calcineurin isn't activated in NPC. Could the depleted acidic organelle Ca²⁺ stores reported in NPC contribute to this since calcineurin is sensitive to changes in lysosomal Ca²⁺ signals?
- FigS4A, PLA could be used here to confirm the absence of an effect on ER-mitochondria MCS.

Reviewer #2 (Remarks to the Author):

The study titled "NPC1-dependent alterations in ER-PM nanodomains drive neurodegeneration" reported loss of NPC1 function elevates KV2.1-CaV1.2 interactions at ER-PM junctions, enhances Ca²⁺ entry into neurons and fuels neurotoxic elevations in mitochondrial Ca²⁺ concentrations. Importantly, targeted disruption of KV2-CaV1 interactions rescues aberrant CaV1.2 clustering, elevated mitochondrial Ca²⁺ and neurotoxicity. This study is very well designed, most data are solid and convincing, also revealed the novel connection that how lysosomal cholesterol may regulate KV2 clustering to tune excitability and Ca²⁺ signaling at ER-PM MCSs to impact neurodegeneration in NPC disease. And the KV2-CaV1 interactions can be a potential therapeutic target in NPC1 disease. Thus, this paper is of interest to the field. However, there is only one experiment (cell viability after NPC1 inhibition) that is directly connected with neurodegeneration. Other are showing various Ca²⁺ alterations inside the cells. Since in NPC disease NPC1 function is altered by mutation during neuronal development it will be necessary to see will be the same effect on mature neurons exist, not only during temporary pharmacological inhibition (48 hrs). So the title appears to be not consistent with the main paper body and key experiments, please correct. In addition, extremely small figures illustration imaging results making reading and assessing the paper very difficult, at some places nearly impossible by human eye. There are also some experimental issues that needs to be addressed. Proposed corrections and questions:

1. Abstract. "NPC disease" should be Niemann-Pick, with brief description of the disease, since it is not very well known by the reader.
2. Figs 1A and 1B. Since values for U18 is lies in the much larger values range figures are hard to read. Probably, making them large or using log scale will helps. This is true for other pictures as well.
3. Fig 3B. Distribution of pictures in counterintuitive. CaV-Kv-merge should lie in the one raw. The same foe other figures. Also there is enough space to make graph bigger.
4. Fig. 3D figures are extremely small, it is nearly impossible to assess the quality and results of experiments. What microscopy (I suppose not confocal) was used for volume analysis should be stated. How images were binarized (or not) to count co-localization?
5. Fig. 6B %SERCA-Kv is really in percent? Or should be multiply by 100?
6. Fig. 7D is repeating neuronal data (which are more important), suggest to put these results in the supplementary.
7. All the ER and Mito Ca image data (Fig7F.G.H.S4B) : since the GCAMP signal intensity not only affected by Ca level, it also affected by expression level, so it is better to show the raw recording trace other than one image. Also in the method part " the cells were recorded basal level in 2 mM Ca²⁺ Ringer's solution, then perfused with a 20 mM Ringer's solution containing 2.5 μM Ionomycin for 6 minutes. The mean intensity ratio of pre-ionomycin/post-ionomycin signals was calculated for each cell. " Could the author specify what is the 20 mM Ringer's solution? Does it have Ca²⁺ in it?
8. Fig7I : it is better to show the represent images for this live/dead assay, also report whether the total cell number changes after treatment, because there is maybe a situation that dead cell detached and can not shown in this assay. especially the n= 992 (CTL), n= 28 (XestoC+U18), why there is so big difference in samples number? Does it because cell detached in this group or the author didn't repeat the experiments properly?
9. SupFigs3B "UA" should be "U18"?
10. TAT-CCAD peptide. How do you control TAT peptide penetration into neurons? How may neurons do uptake DAT peptides? TAT peptides are know to disturb plasma membrane, which are neurons are sensitive to (<https://pubs.acs.org/doi/10.1021/acs.analchem.0c04097>, <https://www.sciencedirect.com/science/article/pii/S0006349509012855> and others) . Since author analyzed plasma membrane protein distribution, it might be crucial. Especially control peptide group is absent on Fig.5. Why control group for CCAD is absent on Fig.5?
11. «Experiments were performed at DIV 6-8, except for the GCaMP3-Kv2.1P404W Ca²⁺ 596 imaging which was conducted at DIV 14-16». But later in the text «Live DIV 13-15 cortical neurons were 787 incubated O/N with Nif, XestoC, Rosc or TAT-CCAD or TAT-Scr» So, for what reason immature neurons were used? And what DIV neurons for other experiments?
12. Author showed in supplementary that ER contacting mitochondria are not changed during NPC1 inhibition with U18. Since ER-PM contacts is enhanced, there is the question whether the basic ER morphology is altered in NPC1 deficient neurons?

Reviewer #3 (Remarks to the Author):

This is an interesting manuscript reporting a series of experimental findings concerning plasma membrane-ER junctions containing Kv2.1 and Cav1.2 and how they are influenced by inhibition of the NPC1 cholesterol transporter. The imaging data appear to be of high quality and the execution of the experiments seems thoughtful and likely to be reliable. Having said that, the vast majority of changes the authors see are quite modest and at least this reader is somewhat uncomfortable with the extent to which they need to trust the authors in order to accept the key conclusions. As discussed below, there are a number of things the authors could do to enhance the rigor within the manuscript.

1) One overarching concern is that the effects the authors measure are quite modest and as far as I could discern, none of the experiments were blinded. This study would seem like a poster child for blinding, and if that had been done the key conclusions would be greatly strengthened. I appreciate that a lot of work went into this study and that blinding all the key experiments is not possible at this late stage. However, would it be possible for the authors to identify a couple of the most critical experiments that could be redone with blinding to enhance the rigor of the study?

2) A second overarching concern is that the U18 inhibitor of NPC1 is used throughout to disrupt NPC1 function. The NPC1 KO cells are used in a few experiments towards the end, but it would considerably strengthen the study if a few other key experiments were reproduced with another means of inhibiting NPC1, perhaps with RNAi or something similar.

3) In general, the figures are packed with data and need to be considerably enlarged to be comprehensible to readers. The population data in Fig 1A,B would be easier to read if plotted as in other figures, but in general it would be ideal if both the images and population data were enlarged and figures split apart, adding additional figures to both the main sections (NC allows 10 main figures) and supplementary data.

4) The methods section should be expanded to include an extensive discussion of how the antibodies used have been validated since many conclusions rest on them being reliable detectors of specific proteins.

5) Quantitation for Fig 5B,C is missing and should be added.

6) In the introduction the authors mention that ER-PM junctions serve a physiologically important role. It would be useful for the reader if the authors could elaborate on this somewhat. My understanding is that the existence of these junctions is fascinating, but that not much is yet known about why they are formed or what functional roles they play.

Comments to the Reviewers

Casas et al.,
NPC1-dependent alterations in Kv2.1–Cav1 nanodomains drive neurodegeneration
NCOMMS-22-33515

We thank the three reviewers for carefully reading our manuscript and providing fair and constructive feedback. We have taken these comments seriously. In the past ~3-4 months we have performed experiments that address all reviewer comments **in full**. New experiments include:

1. Analysis of Kv2.1, Cav1.2, and Kv2.1–Cav1.2 distribution in brain sections from mature NPC1^{I1061T} mice [addresses comments from reviewers 1, 2, 3].
2. Measurement of reactive oxygen species and mitochondrial membrane potential [addresses comments from reviewers 1 and 2].
3. Blinded experiments from key datasets [addresses comments from reviewer 3].
4. Determined upstream roles for mTORC1 and CDK5 [addresses comments from reviewer 1]
5. Tested the role of VAPA/B in helping to organize Kv2.1–Cav1.2 mediated Ca²⁺ entry [addresses comments from reviewer 1].
6. Performed key experiments on NPC1^{-/-} cells [addresses comments from reviewer 2]

Importantly, all new experimental datasets strengthen our overall conclusions and crucially, the rigor of the investigation. It is our hope that given how carefully we addressed each of the reviewer's concerns that our work is now suitable for publication in Nature Communications.

Below (in blue) we provide detailed comments to each individual reviewer. For each comment we provide a figure addressing reviewer comments, i.e. Fig.R1, please note that these figures are for the comments to the reviewer document only, at the end of each comment we provide the location of the new data in the main manuscript file. In the main text we also highlight edits using blue text.

REVIEWER COMMENTS

Reviewer #1 (Remarks to the Author):

This manuscript uncovers a new Ca²⁺ phenotype resulting from NPC1 deficiency. The study elegantly demonstrates clustering of Kv2.1 and Cav1.2 channels at the PM in the absence of functional NPC1, which promotes Ca²⁺ influx to mitochondria via the ER. The findings are novel, timely and of interest to a wide audience. The manuscript is very clearly written, with excellent data presentation (I particularly like the illustrations behind each figure). The findings advance our understanding both of calcium signalling and NPC pathophysiology, but I do have a few comments and concerns:

Response: We thank the reviewer for their kind words and careful reading of our work. Your suggestions for new experiments and editorial changes were fantastic and greatly improved the rigor of the resubmission.

1. The hypothesis is clear and the data support the conclusions but there is a lack of mechanistic detail linking loss of NPC1 function to events at the PM. The authors implicate CDK5 in serine phosphorylation of KV2.1 to increase affiliation with CaV1.2 and promote clustering at the PM which in turn influences PM contact with the ER. How NPC1 function regulates CDK5 activity or how clustering of the channels regulates MCS is not adequately investigated.

Response: We agree. In truth this is something that we had intended to investigate in a subsequent 'follow-up' publication but considering reviewer comments we agree that more evidence should be provided in this publication to better link CDK5 to increased Kv2.1 phosphorylation/clustering (see comments below).

• Is CDK5 upregulated in NPC models (a WB would be nice!)? If so, how? Since mTORC1 is hyperactivated by high cholesterol in NPC, this is a likely candidate (ie, activation of mTORC1 leading to inhibition of AMPK and upregulation of CDK5). Does inhibition of mTORC1 (eg Torin1) have the same effect on U18-mediated clustering as CDK5 inhibition?

Response: Great suggestions.

CDK5 expression: As requested by the reviewer we performed western blots and found that total protein levels of Cyclin-dependent kinase 5 (CDK5) are not significantly different between control and NPC1 loss of function cells (**Fig.R1A**). Given published work demonstrating that CDK5 activity is increased in NPC disease¹, and our new experiments that phosphorylation of Kv2.1 S603 (a direct target of CDK5²) is increased following loss of NPC1 function (see Fig.R3A, comments below) we focused our attention on how CDK5 activity could be increased following NPC1 loss of function. We performed western blot experiments for two CDK5 activators, p35 and p39 and determined that both exhibit ~2.5-fold increases in total protein expression relative to control (**Fig.R1A**). Given that CDK5 is recruited to the plasma membrane through p35-dependent myristoylation^{3,4} we also performed double-immunolabel single molecule localization microscopy (SMLM) for Kv2.1 and CDK5 and found that the fractional amount of CDK5 overlapping with Kv2.1 (at the 20 nm lateral resolution of the system⁵) increased when NPC1 was inhibited (**Fig.R1B**). These experiments support a model where CDK5-mediated phosphorylation of Kv2.1 at residue S603 is enhanced due to increased CDK5 activation and anchoring near Kv2.1 (through increased p35/39) at the plasma membrane. This phosphorylation promotes increase clustering of Kv2.1 and Cav1.2–Kv2.1. Lending further credence to this model, pharmacological blockade of CDK5 activity using Roscovitine (Rosco), rescued all cellular phenotypes including Kv2.1 clustering, Cav1.2 clustering, Cav1.2–Kv2.1 clustering (**Fig.R1C**), elevations in Mito Ca²⁺ (**Fig.R1D**) and cellular toxicity (**Fig.R1E**). These data support that CDK5, and its activating proteins, are key upstream mediators

of NPC1-dependent toxicity. These new experiments collectively presented below in **Fig.R1** and can be found in Figs. 5C, 5D, 6; 10F and S9A.

Fig.R1. CDK5 inhibition decreases *K_v2.1*–*Ca_v1.2* distribution and activity in a model of NPC disease. **(A)** Top: Representative western blots of CDK5, p35, p39, and GAPDH under control (CTL) and NPC1-inhibition (U18) conditions. Bottom:

quantification of changes in protein expression relative to CTL. **(B)** Left: representative super resolution localization map of CDK5 (1st column), $K_V2.1$ (2nd column) and Merge (3rd column) of CDK5 and $K_V2.1$ at the plasma membrane (PM) under CTL and U18 conditions. Right: quantification showing % of the soma occupied by $K_V2.1$, CDK5, and $K_V2.1$ –CDK5 complexes. **(C)** Top: representative super resolution localization map of $Ca_v1.2$ (1st row), $K_V2.1$ (2nd row), and Merge (3rd row) of $Ca_v1.2$ and $K_V2.1$ to the PM under CTL, U18, Roscovitine, and U18 + Roscovitine conditions. Bottom: quantification showing % of the soma occupied by $K_V2.1$ – $Ca_v1.2$ complexes. **(D)** Top: representative super resolution mitochondrial calcium imaging under CTL and U18 conditions. Bottom: quantification. **(E)** Top: representative images of cell viability assay under CTL, U18, Roscovitine, and U18 + Roscovitine conditions. Bottom: quantification.

Inhibition of mTORC1: As noted by the reviewer, mTORC1 is hyperactive following loss of NPC1 function⁶. To test a role for mTORC1 in mediating/influencing $K_V2.1$ – $Ca_v1.2$ channel interactions we inhibited mTORC1 with the pharmacological inhibitor Torin-1 and performed double-immunolabel SMLM imaging for $K_V2.1$ and $Ca_v1.2$ with and without inhibition of NPC1. Consistent with a role for mTORC1 in linking NPC1 loss-of-function to enhanced $K_V2.1$ – $Ca_v1.2$ clustering, inhibition of mTORC1 abrogated U18-dependent increases in $K_V2.1$ – $Ca_v1.2$ (**Fig.R2**). The precise molecular mechanism(s) through which this occurs remains to be determined but is beyond the scope of the present study. We have included these data in the updated manuscript and updated the discussion to suggest future investigations target ERK, AMPK, and SREBP. What is clear is that both mTORC1 and CDK5/p35 are upstream regulators of $K_V2.1$ – $Ca_v1.2$ and targeting either can reduce NPC1 toxicity. These new data appear in the manuscript in Fig. S5B.

Fig.R2. *Inhibition of mTORC1 abrogates NPC1-mediated increases in K_v2.1–Ca_v1.2. Left: representative super resolution images of Cav1.2 (1st row), K_v2.1 (2nd row), and Merge (3rd row) of Cav1.2 and K_v2.1 to the PM under CTL, U18, Torin-1 and U18 + Torin-1 conditions. Right: quantification showing % of the soma occupied by K_v2.1–Ca_v1.2 complexes.*

• Does S603 phosphorylation affect VAP binding or is this effect on MCS independent of K_v2.1–VAP interaction? S603 is approx 10a/a downstream of K_v2.1's noncanonical FFAT motif. Is K_v2.1–VAP interaction required for the effects on Ca²⁺ signalling - do you still see Ca²⁺ influx in cells depleted of VAPs?

Response: Thanks for these suggestions. We tested if both S603 phosphorylation status and VAP binding are altered following loss of NPC1 function.

1. **K_v2.1 S603:** As noted above, we performed super-resolution imaging using an antibody that recognizes phosphorylation at S603 (pS603²) and determined that following loss of NPC1 function there is a significant increase in the number of pS603 puncta (**Fig.R3A**). As discussed above, S603 is a substrate for CDK5, thus the increase in CDK5-activating partners and its enhanced anchoring in proximity to K_v2.1 aligns well with increases in pS603 and suggests that increased CDK5 activity promotes enhanced K_v2.1 clustering. Additionally, increases in pS603 were mirrored by increases in VAPA/B clustering (**Fig.R3B**) and K_v2.1–VAPA/B overlap (**Fig.R3B**) within neuronal TIRF footprints suggesting that K_v2.1 is phosphorylated at multiple sites following inhibition of NPC1 cholesterol transport.
2. **K_v2.1–VAP and Ca²⁺ signaling:** to test if K_v2.1–VAP interactions are required for the effects on Ca²⁺ signaling we expressed GCaMP-K_v2.1_{P404W} (non-conducting K_v2.1 that localizes to ER–PM K_v2.1 contact sites) and treated NPC1-inhibited neurons with a peptide (TAT-FFAT) that inhibits VAP binding to K_v2.1 by blocking the FFAT binding domain in VAPA/B proteins and thereby disrupting the association between K_v2.1 and VAPA⁷. A scrambled peptide sequence was used as a control. Subsequent high-speed TIRF imaging revealed that treatment of neurons with the TAT-FFAT peptide abrogated increases in cytoplasmic Ca²⁺ near K_v2.1 channels (**Fig.R3C**). Datasets with the FFAT peptide mirror results with the CCAD peptide (**Fig.R3C**) and presents a model whereby collaborative K_v2.1–Ca_v1.2–VAPA/B interactions are required for elevated cytoplasmic Ca²⁺ entry in NPC disease. These new data appear in Figs. 5B, 7B and 8C with comments included in the discussion section.

Fig.R3. Role of pS603 and FFAT in K_v2.1–Ca_v1.2 Ca²⁺ signaling in NPC disease. **(A)** Left: representative super resolution localization map of K_v2.1 pS603 under CTL and U18 conditions. Right: quantification. **(B)** Left: representative super resolution localization map of VAPA/B (1st column), K_v2.1 (2nd column) and Merge (3rd column) of VAPA/B and K_v2.1 to the PM under CTL and U18 conditions. Right:

quantification showing % of the soma occupied by Kv2.1, VAPA/B, and overlapping Kv2.1–VAPA/B complexes. (C) Left: Representative neurons expressing GCaMP-Kv2.1_{P404W} and calcium traces under CTL, U18, U18 + CCAD peptide, or U18 + FFAT peptide conditions. Right: analysis of intensity changes in GCaMP-Kv2.1_{P404W}.

Do you see evidence of mitochondrial dysfunction(eg. mtROS, membrane potential etc) in your models of NPC and if so can you relate it to the channel clustering - ie, does TAT-CCAD rescue mitochondrial dysfunction?

Response: Thank you for this important comment. In new experiments to directly answer this question we measured intracellular ROS and mitochondrial membrane potential in models of NPC1 disease and found that loss of NPC1 function resulted in a more depolarized mitochondrial membrane potential and increased ROS production. Crucially, co-treating neurons with the TAT-CCAD (CCAD) peptide rescued changes in mitochondrial membrane potential and ROS production suggesting that Kv2.1–Cav1.2 interactions are upstream drivers of mitochondrial dysfunction. These data are presented below in **Fig.R4** and in Figs.10D and 10E of the main text.

Fig.R4. Targeting Kv2.1–Cav1.2 interactions with CCAD peptide rescues ROS and mitochondrial membrane potential in a model of NPC disease. (A) Representative confocal images of ROS aggregates under CTL (TAT-HA-ClaB-Scr peptide, SCRBL), U18, CCAD peptide, and U18 + CCAD conditions. Right: analysis of ROS aggregates. (B) Quantification of mitochondrial membrane potential.

3. Thapsigargin is used to inhibit SERCA-mediated Ca²⁺ sequestration into the ER, preventing elevations in mitochondrial Ca²⁺ which are proposed to result from the increased channel clustering at the PM. However, since SERCA inhibition also results in increased cytosolic [Ca²⁺], wouldn't thapsigargin be predicted to promote mitochondrial calcium uptake? If including the thapsigargin data, this needs a bit of explanation - could thapsigargin be activating calcineurin-mediated Kv2.1 dephosphorylation to downregulate clustering etc? It might be over-complicating things but could fairly easily be checked with anti-pS603?

Response: To address this comment we fixed and labeled neurons for K_v2.1 pS603 following treatment with U18 or Thapsigargin (to inhibit SERCA). The data below underscores the importance of the reviewers' comments as Thapsigargin alone decreased (albeit not significantly) K_v2.1 pS603 (**Fig.R5**). As suggested by the reviewer this is likely due to calcineurin-dependent dephosphorylation at K_v2.1 pS603. However, when neurons were co-treated with both U18 and Thapsigargin there was still an U18-dependent increase in K_v2.1 pS603 (**Fig.R5**). Thus, it appears that experiments involving Thapsigargin treatment represent a convolution of effects and as such do not allow us to clearly state that a single definitive factor i.e. preventing elevations in mito Ca²⁺ through ER Ca²⁺ depletion or declustering of K_v2.1 is responsible for improving cell viability in NPC models of disease. It is for this reason that we are removing the Thapsigargin datasets from the cell viability experiments. It is important to note that removal of this dataset does not impact the overall conclusions of our work.

Fig.R5. K_v2.1 pS603 phosphorylation status during NPC1 and/or SERCA inhibition. Left: representative super resolution localization map of K_v2.1 pS603 to the PM of neurons under CTL, thapsigargin (Thaps), U18, and U18 + Thaps, fixed and labeled for K_v2.1 pS603 (L61/14.2). Right: quantification.

4. Does inhibition of clustering-mediated calcium influx (eg inhibition of Ca_v1 or IP₃R) prevent higher frequency APs? I don't know how involved these experiments are - would make a nice addition but the story is also strong without!

Response: To test this suggestion we expressed a genetically encoded voltage indicator (ArchLight-Q239) in neurons and treated with or without U18 and CCAD (to uncouple KV2.1-CaV1 calcium entry). Analysis of ArchLight-Q239 intensity changes revealed that U18-treated neurons have a hyperexcitable membrane potential compared to CTL irrespective of whether cells were treated with CCAD (**Fig.R6**). These data are included in the main text in Fig. S7B

Fig.R6. NPC1-dependent hyperexcitability is upstream of increases in Cav1.2–Kv2.1 interaction. Top: representative TIRF footprints of neurons expressing ArchLight-Q239 and normalized traces under CTL, U18, CCAD and U18 + CCAD peptide conditions. Bottom: quantification.

5. Minor points:

- There is possibly an over-reliance on NPC1 inhibition by U18666A, which has been reported to have off-target effects. However, validation both of some data and the use of U18 by results obtained from other models of NPC1 deficiency provides a high level of confidence in the findings.

Response: Thank you for this comment. In the resubmission we have included more datasets from the NPC1^{I1061T} animals and also from NPC1^{-/-} cells. We have detailed these datasets in our responses to reviewer 2. Importantly, these new datasets confirm the results previously presented from U18 treatment. It should be noted that this has been our experience across previously published studies⁸⁻¹¹.

- p13, line 351 says that results demonstrate a role for NPC1 in organizing coupling between channels etc. This is slightly misleading as, unless I have misunderstood, NPC1 is not directly involved in organizing the channels. Maybe it could be softened to demonstrating novel downstream effects of NPC1 function (NPC1-mediated lipid transport?) in regulating the organization of channels at the PM?

Response: Thank you for this important note. We have updated the text to ensure readers are aware that any NPC1-dependent reorganization is occurring due to downstream lipid-dependent remodeling.

- p13, line 419 Thapsigargin typo.

Response: Thank you, corrected.

- p18, line 494 questions why calcineurin isn't activated in NPC. Could the depleted acidic organelle Ca²⁺ stores reported in NPC contribute to this since calcineurin is sensitive to changes in lysosomal Ca²⁺ signals?

Response: The reviewer raises an interesting point which we have now included in the discussion.

- FigS4A, PLA could be used here to confirm the absence of an effect on ER-mitochondria MCS.

Response: In new Fig. S8A (old Fig. S4A) we are looking at gross changes in ER-Mito contact sites using super-resolution AiryScan imaging. As noted by the reviewer we could have also used PLA to test if ER-Mito membrane contacts are grossly perturbed following NPC1 inhibition. To do this we would have to select two stochastically distributed and abundant proteins and confirm that their abundance is not dependent on NPC1 transport function. Given that we already detail molecular changes between IP₃R1, GRP75, and VDAC1 which are at ER-Mito contact sites we think these additional experiments would be redundant within the context of our results.

Reviewer #2 (Remarks to the Author):

The study titled “ NPC1-dependent alterations in ER–PM nanodomains drive neurodegeneration” reported loss of NPC1 function elevates KV2.1–CaV1.2 interactions at ER–PM junctions, enhances Ca²⁺ entry into neurons and fuels neurotoxic elevations in mitochondrial Ca²⁺ concentrations. Importantly, targeted disruption of KV2–CaV1 interactions rescues aberrant CaV1.2 clustering, elevated mitochondrial Ca²⁺ and neurotoxicity. This study is very well designed, most data are solid and convincing, also revealed the novel connection that how lysosomal cholesterol may regulate KV2 clustering to tune excitability and Ca²⁺ signaling at ER–PM MCSs to impact neurodegeneration in NPC disease. And the KV2–CaV1 interactions can be a potential therapeutic target in NPC1 disease. Thus, this paper is of interest to the field.

Response: Thank you for your careful reading of the manuscript, your kind words, and constructive feedback. We appreciate it. Below we answer your comments/suggestions in full.

However, there is only one experiment (cell viability after NPC1 inhibition) that is directly connected with neurodegeneration. Other are showing various Ca²⁺ alterations inside the cells. Since in NPC disease NPC1 function is altered by mutation during neuronal development it will be necessary to see will be the same effect on mature neurons exist, not only during temporary pharmacological inhibition (48 hrs). So the title appears to be not consistent with the main paper body and key experiments, please correct.

Response: Thank you for this important comment. We begin by noting that across multiple published studies we have always conducted experiments from cells/neurons, including mature neurons, that have disease mutations in NPC1, NPC1 knocked out, and NPC1 inhibited with the U18666A (U18) compound. Every experiment, from each model of NPC disease has produced identical results⁸⁻¹¹. Thus, this supports that the NPC1 inhibitor, U18, is a faithful inhibitor of NPC1 and an excellent model of NPC disease. That said, based on review comments, we have repeated several experimental datasets in mature neurons from NPC1^{I1061T} mice and show that like U18 treatment of isolated neurons we see similar increases in K_v2.1 and Ca_v1.2 clustering. Specifically, we fixed, sectioned, and immunolabeled brain slices from 60-day old age and sex-matched WT or NPC1^{I1061T} mice and performed multiplexed immunofluorescence imaging for K_v2.1, Ca_v1.2, Hoechst and/or Calbindin. First, we quantified K_v2.1 distribution in the cortex and found that like isolated ‘immature’ neurons from this region, K_v2.1 clusters were significantly larger in NPC1^{I1061T} relative to WT mice (see below, **Fig.R7**); these datasets now make up a revised Fig.3 in the main text. Like the cortex, the CA1 region of the NPC1^{I1061T} hippocampus also exhibited increases in K_v2.1 cluster size relative to WT control (**Fig.R8**). These data can be found in an updated Fig. S3. Finally, we performed experiments in the cerebellum of WT and NPC1^{I1061T} animals as Purkinje cells have been demonstrated to be vulnerable in NPC disease. Like the cortex and CA1 region of the hippocampus, Calbindin-positive Purkinje cells showed increases in K_v2.1 and Ca_v1.2 cluster size (**Fig.R9**) as well as overlapping volume between the two proteins. This new dataset is included in an updated Fig.4C in the main text.

Fig.R7. NPC1 inhibition or $NPC1^{H1061T}$ disease mutation increase $K_v2.1$ cluster distribution. **(A)** Left: representative super resolution localization map of $K_v2.1$ to the plasma membrane (PM) under control (CTL), NPC1 inhibition (U18). Right: quantification. **(B)** Left: representative super resolution TIRF footprints of CTL and U18 neurons fixed and labeled for $K_v2.1$. Right: quantification. **(C)** Cortical brain

sections from WT (left) or NPC1^{I1061T} (right) animals stained for Kv2.1 (1st row) and Hoechst (2nd row). Right: quantification.

Fig.R8. NPC1^{I1061T} disease mutation increase Kv2.1 cluster distribution in the CA1 region of hippocampus. Kv2.1 (1st row) or Hoechst (2nd row) labeling from the CA1 region of WT or NPC1^{I1061T} hippocampi.

Fig.R9. NPC1^{I1061T} disease mutation increases Kv2.1 and Cav1.2 cluster distribution in Purkinje cells of the cerebellum. Left: multiplexed Calbindin (1st row), Cav1.2 (2nd row), Kv2.1 (3rd row) and Merge (4th row) of Cav1.2 and Kv2.1 imaging from WT (left) and NPC1^{I1061T} (right) cerebella. Right: quantification of Cav1.2-Kv2.1 overlap.

In addition to new experiments brain sections from adult mice, we have also performed additional experiments using NPC1^{-/-} cells exogenously expressing Cav1.2 and Kv2.1. As can be seen in **Fig.R10** (below) and Fig. S4 of the main text, NPC1 knockout increases the size of both Kv2.1 and Cav1.2 clusters, as well as the % overlap between the two proteins. Furthermore, treating neurons with the CCAD peptide normalizes all increases back to control levels. Therefore, similar to U18-treated or NPC1^{11061T} neurons, knocking out NPC1 increases Kv2.1–Cav1.2 nanocomplexes.

Fig.R10. NPC1^{-/-} cells have increased Kv2.1 and Cav1.2 clustering. Left: representative super resolution localization map of Cav1.2 (1st row), Kv2.1 (2nd row), and Merge of Cav1.2 and Kv2.1 to the PM (3rd row) in WT and NPC1^{-/-} under conditions with or without TAT-HA-C1aB-Scr (SCRBL) or TAT-HA-C1aB (CCAD) peptides. Right: quantification. Note that CCAD abrogates NPC1-dependent increases in Kv2.1 and Cav1.2. Note that % refers to the % of the soma occupied by Kv2.1–Cav1.2 complexes.

In addition, extremely small figures illustration imaging results making reading and assessing the paper very difficult, at some places nearly impossible by human eye.

Response: We apologize for the small figure panels in our original submission. We have now expanded each of the figures to make panels larger. With figure expansion and inclusion of new datasets the paper has grown to 10 main figures and 10 supplement figures.

There are also some experimental issues that needs to be addressed. Proposed corrections and questions:

1. Abstract. "NPC disease" should be Niemann-Pick, with brief description of the disease, since it is not very well known by the reader.

Response: Thank you for this comment. We have included a brief description in the abstract to better introduce NPC disease.

2. Figs 1A and 1B. Since values for U18 is lies in the much larger values range figures are hard to read. Probably, making them large or using log scale will helps. This is true for other pictures as well.

Response: Thank you for this comment. We apologize for the difficulty in observing datasets. We have updated all presentations to make the data easier to visualize. Please see an example from Fig.1C, D below.

3. Fig 3B. Distribution of pictures in counterintuitive. CaV-Kv-merge should lie in the one row. The same foe other figures. Also there is enough space to make graph bigger.

Response: We agree with the reviewer and have reorganized the figures as suggested to have the merged image alongside the raw data and enlarged the histograms to make them easier to visualize. Please see the example from the new Fig.4A (formerly Fig.3B below).

4. Fig. 3D figures are extremely small, it is nearly impossible to assess the quality and results of experiments. What microscopy (I suppose not confocal) was used for volume analysis should be stated. How images were binarized (or not) to count co-localization?

Response: We apologize for the small size of figure and lack of details regarding experimental conditions. Images of brain sections were taken from P60 animals using a Zeiss ApoTome. In the original submission we did binarize image stacks to generate 3-D reconstructions however instead of presenting in black and white like Fig S8A we pseudocolored them to match the raw images. In the resubmission we have taken new images, reorganized the image presentation, increased the size of the images. See example below from Fig.4D (formerly Fig.3D).

5. Fig. 6B %SERCA-Kv is really in percent? Or should be multiply by 100?

Response: Thank you for this comment. Yes, the value of SERCA-K_v2.1 overlap is 0.015 % in Control and 0.025 % in U18. Why do these values appear low? Both SERCA and K_v2.1 images (technically localization maps) were generated using a super-resolution STORM microscope that has a resolution of 20 nm^{5,12}. Of course, such enhanced resolution compared to confocal microscopy means that the probability of two signals being close to one another decreases. Our analysis (%) refers to the amount of SERCA-K_v2.1 occupying the total area of the neuron soma. In this way 0.025 % means that the overlapping SERCA-K_v2.1 signal occupies 1/4,000 of the total neuron soma area.

6. Fig. 7D is repeating neuronal data (which are more important), suggest to put these results in the supplementary.

Response: Thank you for this comment. We agree and have moved this dataset to the supplement.

7. All the ER and Mito Ca image data (Fig7F.G.H.S4B) : since the GCAMP signal intensity not only affected by Ca level, it also affected by expression level, so it is better to show

the raw recording trace other than one image. Also in the method part “ the cells were recorded basal level in 2 mM Ca²⁺ Ringer’s solution, then perfused with a 20 mM Ringer’s solution containing 2.5 μM Ionomycin for 6 minutes. The mean intensity ratio of pre-ionomycin/post-ionomycin signals was calculated for each cell. “ Could the author specify what is the 20 mM Ringer’s solution? Does it have Ca²⁺ in it?

Response: Thank you for this comment. 20 mM Ringers solution does refer to the solution containing 20 mM Ca²⁺. The complete solution is (in mM): 160 NaCl, 2.5 KCl, 20 CaCl₂, 1 MgCl₂, 10 HEPES, and 8 D-Glucose. In the resubmission we include representative raw intensity traces for pCAG-mito-RCaMP1h (mito Ca²⁺) and ER-GCaMP6-150 (ER Ca²⁺) (see below; **Fig.R11**). As the reviewer can see the resting mitochondrial RCaMP1h intensity values are significantly greater in U18 treated cells while ER GCaMP6 intensities are significantly reduced. These raw intensity datasets support the conclusion that inhibition of NPC1 with U18 in neurons increases mitochondrial Ca²⁺ and decreases ER Ca²⁺. These data now appear in Fig. S9B and S9C.

Fig.R11. Loss of NPC1 function increases mitochondrial Ca²⁺ while reducing ER Ca²⁺. **(A)** Top: representative images of neurons expressing Mito-RCaMP1h under CTL and U18 conditions, with or without ionomycin. Bottom: Time course of increase in Mito-RCaMP1h intensity following application of ionomycin. **(B)** Same as **(A)** but expressing ER-GCaMP6-150 instead.

8. Fig7I : it is better to show the represent images for this live/dead assay, also report whether the total cell number changes after treatment, because there is maybe a situation that dead cell detached and can not shown in this assay. especially the n= 992 (CTL), n=

28 (XestoC+U18), why there is so big difference in samples number? Does it because cell detached in this group or the author didn't repeat the experiments properly?

Response: As suggested by the reviewer, we have now included the representative images from the live/dead cell experiments together with the fold change of each condition respective to their controls (**Fig.R12** below). This dataset now appears in Fig. 10F of the main text.

Fig.R12. Targeting $Kv2.1-Cav1$ interactions rescues neurotoxicity in NPC disease. Quantification of neuronal viability as a ratio between live cell/dead cell normalized to their respective controls. $N= 394$ (DMSO), 174 (U18), $n= 94$ (XestoC), $n= 28$ (XestoC + U18), $n= 124$ (SCRBL), $n= 50$ (SCRBL + U18), $n= 85$ (CCAD), $n= 59$ (CCAD + U18), $n= 33$ (Rosc) and $n= 29$ (Rosc + U18) neurons were analyzed.

The reviewer also raises an important point about the number of cells for each dataset. In our original submission we aggregated control datasets into a single pool i.e. 'without U18' contained normalized values for DMSO, Xesto C alone, Nifedipine alone, and Roscovitine alone. We did so because we wanted to visualize the U18-dependent changes in viability while having the minimum number of histogram plots (for clarity). We realize now, based on the reviewer comments, that this is confusing. Therefore, in the resubmission we plot the fold change of each condition (compared to their respective controls) and analyze each treatment effect as any significant change different from "1". In agreement with the reviewer comment regarding cellular attachment, we also wanted to show the cell density for each condition. (**Fig.R13** below or Fig 9D in the current manuscript).

Fig.R13. Targeting $Kv2.1-Cav1.2$ interactions rescues cellular density in NPC disease. Quantification of cellular density. $N= 41$ (DMSO), $n= 55$ (U18), $n= 12$

(XestoC + U18), n= 27 (Rosco + U18), n= 13 (SCRBL + U18), n= 14 (CCAD + U18) imaging field areas were imaged and analyzed.

As the reviewer can observe, some of the treatments led to cellular detachment, which could explain reduced data points for some of the conditions in the “Viability fold change” histogram. Importantly, the number of images used is similar for each condition and the conclusions of the data are the same, disruption of the $K_v2.1$ – $Ca_v1.2$ interaction significantly improves both cell viability and density in NPC1-inhibited neurons.

9. SupFigs3B “UA” should be “U18”?

Response: Thank you. We have edited the figure accordingly.

10. TAT-CCAD peptide. How do you control TAT peptide penetration into neurons? How may neurons do uptake DAT peptides? TAT peptides are known to disturb plasma membrane, which neurons are sensitive to (<https://pubs.acs.org/doi/10.1021/acs.analchem.0c04097>, <https://www.sciencedirect.com/science/article/pii/S0006349509012855> and others). Since author analyzed plasma membrane protein distribution, it might be crucial. Especially control peptide group is absent on Fig.5. Why control group for CCAD is absent on Fig.5?

Response: Thank you for this important comment. As a control we treat neurons with a scrambled peptide that also contains a TAT domain. We have edited our methods section to reflect this important consideration. Additionally, for all our experiments we counterstain for anti-HA to ensure presence of the TAT-HA-C1aB-Scr (SCRBL) and TAT-HA-C1aB (CCAD) peptides in neurons. We include representative images of neurons treated with SCRBL and CCAD peptides and fixed and labeled for anti-HA (**Fig. R14**).

Fig.R14. Targeting $Kv2.1-Cav1$ with CCAD peptide abrogates increases in $Cav1.2$ clustering. Left: representative super resolution localization map of $Cav1.2$ in neurons under SCRBL, SCRBL + U18, CCAD, or CCAD + U18, and fixed and stained for $Cav1.2$ and HA. Note inverted grayscale inserts depict presence of either TAT-HA-ClaB-Scr (SCRBL) or TAT-HA-C1aB (CCAD). Right: quantification.

Furthermore, we now included all datasets from the former Fig.5 (**Fig.R15**, see below). These data illustrate that targeting $Kv2.1-Cav1.2$ interactions with CCAD rescue NPC1-independent increases in ER-PM junction. These data appear in the updated Fig.7 in the main text.

Fig.R15. Targeting $Kv2.1-Cav1.2$ with CCAD peptide abrogates NPC1-dependent increases in ER-PM membrane contacts. Left: representative images from neurons expressing MAPPER in SCRBL, SCRBL + U18, CCAD, and CCAD + U18. Right: quantification.

11. «Experiments were performed at DIV 6-8, except for the GCaMP3-Kv2.1P404W Ca^{2+} 596 imaging which was conducted at DIV 14-16». But later in the text «Live DIV 13-15 cortical neurons were 787 incubated O/N with Nif, XestoC, Rosc or TAT-CCAD or TAT-Scr» So, for what reason immature neurons were used? And what DIV neurons for other experiments?

Response: GCaMP3-Kv2.1P404W Ca^{2+} imaging was performed at DIV 14-16 to allow for the appropriate transfection efficacy of the plasmid^{13,14}, i.e. neurons were transfected at DIV 12-14, incubated for 2 days to allow appropriate expression of GCaMP3-Kv2.1P404W, and then treated overnight with U18 at DIV 13-15 before imaging at DIV 14-16. We

selected DIV 6-8 for all the other fluorescence experiments because similar immunofluorescence distributions were observed from DIV6-8 through DIV21, better expression of MAPPER, pCAG-mito-RCamP1h, ER-GCaMP6-150, SPLICS_{L/S}-P2A^{ER-PM} and ArchLight-Q239 plasmids was shown at this earlier time point, and Live/Dead ratios were similar across all time points. We apologize for having noted DIV 14-16 in the Viability Assay Methodology section in the former manuscript, we have now corrected this in the revised manuscript. It should also be noted that new datasets from mature WT and NPC1^{I1061T} P60 animals have similar trends for Kv2.1, Cav1.2, and Kv2.1–Cav1.2 as isolated neurons from control and U18-treated animals. Finally, we have shown on multiple occasions that inhibition of NPC1 with U18 elicits identical responses as those observed in mature neurons ^{8,9,11,15}.

12. Author showed in supplementary that ER contacting mitochondria are not changed during NPC1 inhibition with U18. Since ER-PM contacts is enhanced, there is the question whether the basic ER morphology is altered in NPC1 deficient neurons?

Response: Thank you for this comment. We have subsequently tested if basic ER morphology is altered in NPC1 deficient neurons by performing a volumetric analysis using live-neuron ER dye tracker. As shown below (**Fig.R16**) and in new Fig. S6, the basic morphology of the ER doesn't change at this resolution, rather as we show throughout the manuscript it is discrete contacts between the PM and ER that increase and appear to be important drivers of neurodegeneration.

Fig.R16. Gross ER morphology does not seem to be altered by U18 treatment. Left: representative maximum intensity projections of neurons loaded with the ER dye under CTL and U18 conditions. Right: quantification.

Reviewer #3 (Remarks to the Author):

This is an interesting manuscript reporting a series of experimental findings concerning plasma membrane-ER junctions containing $K_v2.1$ and $Ca_v1.2$ and how they are influenced by inhibition of the NPC1 cholesterol transporter. The imaging data appear to be of high quality and the execution of the experiments seems thoughtful and likely to be reliable. Having said that, the vast majority of changes the authors see are quite modest and at least this reader is somewhat uncomfortable with the extent to which they need to trust the authors in order to accept the key conclusions. As discussed below, there are a number of things the authors could do to enhance the rigor within the manuscript.

Response: Thank you for your important and thoughtful comments. We have tried to address these comments in full. We would like to note that ion channels exert incredible influence over neuronal physiology, with minor alterations in their function capable of causing large changes in the membrane potential and /or exert profound downstream effects. For example, in an excerpt taken from the Molecular Biology of the Cell they note that *“a spherical cell of diameter 10 μm , the number of K^+ ions that have to flow out to alter the membrane potential by 100 mV is only about 1/100,000 of the total number of K^+ ions in the cytosol”*. The changes we see are clearly much greater than 0.00001 % noted above yet also exert incredible influence over neuronal pathophysiology. This said, the reviewer raises an important point about the perception of small changes and their importance in the context of something much larger, i.e., a neuron undergoing neurodegeneration. We begin addressing this question below by performing blinded experiments and determined that the changes observed when experiments were blinded were larger! In addition, we have addressed all the reviewers' comments in full to increase the rigor of the manuscript.

1) One overarching concern is that the effects the authors measure are quite modest and as far as I could discern, none of the experiments were blinded. This study would seem like a poster child for blinding, and if that had been done the key conclusions would be greatly strengthened. I appreciate that a lot of work went into this study and that blinding all the key experiments is not possible at this late stage. However, would it be possible for the authors to identify a couple of the most critical experiments that could be redone with blinding to enhance the rigor of the study?

Response: Thank you. This was an excellent suggestion. In response to reviewer comments, we have completed two critical blinded datasets that represent the most critical experiments in the manuscript. We conducted the following blinded experiments:

1. Super resolution imaging of $K_v2.1$ and $Ca_v1.2$ in neurons (**Fig.R17** below and **Fig.S4A** in the main manuscript)

Fig.R17. Blinded experiments: NPC1 inhibition increases Cav1.2 clustering, Kv2.1 clustering and Cav1.2-Kv2.1 proximity. Top: representative super resolution localization map of Cav1.2 and Kv2.1 localized to the plasma membrane (PM) under control (CTL) and NPC1-inhibition (U18) conditions. Bottom: quantification.

2. Super resolution imaging of IP₃R1 and VDAC1 in neurons (**Fig.R18** below and Fig.S8E of main manuscript).

Fig.R18. Blinded experiments: NPC1 inhibition increases IP₃R–VDAC1 proximity. Top: representative super resolution localization map of IP₃R and VDAC1 under CTL and U18 conditions. Bottom: quantification.

As the reviewer can see from the data above, we also show in blinded experiments that loss of NPC1 function with U18 resulted in increases in cluster sizes and overlap between K_v2.1–Ca_v1.2 and IP₃R1–VDAC1. We believe these datasets increase the overall rigor of our investigations.

2) A second overarching concern is that the U18 inhibitor of NPC1 is used throughout to disrupt NPC1 function. The NPC1 KO cells are used in a few experiments towards the end, but it would considerably strengthen the study if a few other key experiments were reproduced with another means of inhibiting NPC1, perhaps with RNAi or something similar.

Response: Thank you for this comment. Across several published studies we have consistently compared U18, NPC1^{-/-}, and NPC1^{I1061T} and in every investigation all each

treatment group has yielded identical results. That said, to increase the rigor of our investigations we have added additional datasets using NPC1^{-/-} cells and conducted experiments from NPC1^{I1061T} mice. These datasets can be observed in comments to reviewers 1 and 2. As in our previous publications, the results using NPC1 inhibition (U18) are identical to those with NPC1^{-/-} and NPC1^{I1061T}. Specifically we have conducted the following experiments for our resubmission:

1. Shown and quantified Cav1.2 and Kv2.1 distribution in WT and NPC1^{I1061T} brain sections (See **Fig. R7, R8, and R9**)
2. Quantified Kv2.1–Cav1.2 distribution in WT and NPC1^{I1061T} brain sections (**Fig.R9**)
3. Quantified Kv2.1–Cav1.2 in WT and NPC1^{-/-} cells incubated with a scrambled or CCAD peptide (**Fig. R10**)
4. Quantified IP₃R1–VDAC in Control and NPC1^{-/-} cells (**Fig.R19** below and Fig. S8C)

Importantly, all datasets with U18 matched those from NPC1^{I1061T} and NPC1^{-/-} cells.

Fig.R19. Increased IP₃R–VDAC1 proximity in NPC1^{-/-} cells. Top: representative images of WT and NPC1^{-/-} cells fixed and labeled for IP₃R and VDAC1. Bottom: quantification.

3) In general, the figures are packed with data and need to be considerably enlarged to be comprehensible to readers. The population data in Fig 1A,B would be easier to read if plotted as in other figures, but in general it would be ideal if both the images and population data were enlarged and figures split apart, adding additional figures to both the main sections (NC allows 10 main figures) and supplementary data.

Response: Thank you. We agree. As noted to reviewer 2, we have enlarged and expanded our figures. In the first submission there were seven figures, which we have increased to ten in the resubmission. Additionally, the supplement has expanded to 10 figures. We believe this now increases accessibility of our data to a wider audience.

4) The methods section should be expanded to include an extensive discussion of how the antibodies used have been validated since many conclusions rest on them being reliable detectors of specific proteins.

Response: Thank you. We agree. Dr. Trimmer and his group are world renowned for their antibody development and have strict protocols in place to test each antibody. In fact Dr. Trimmer founded the UC Davis/NIH NeuroMab Facility (<https://neuromab.ucdavis.edu>), and several of the key antibodies we used were developed at that facility. Each of these antibodies has been validated on knockout and/or knockdown samples. Such strict protocols (immunohistochemistry, western blot, knockout cells/brains, overexpression) have meant that many of these antibodies have been used in published studies many times and thus are extremely well characterized. As an example, we list several of the antibodies used in the manuscript below and note the number (likely underestimation) of citations each antibody has received based on CiteAb. We have made this clear in the methods section.

Protein	Antibody	Citations
Kv2.1	K89/34; NeuroMab	82
Cav1.2	ACC-003; Alomone	208
VDAC1	ab14734; Abcam	595

5) Quantitation for Fig 5B,C is missing and should be added.

Response: Thank you. We have now included the quantification for the former Fig.5B,C. These data now appear in Fig.7 of the main text and can be seen in **Fig.R20**.

Fig.R20. Increased ER-PM junctions under NPC1 inhibition. **(A)** Left: representative images of neurons transfected with the short SPLICS-P2A^{ER-PM} (10 nm) under CTL and U18 conditions. Right: quantification. **(B)** Same as (A) but using the long SPLICS-P2A^{ER-PM} (40 nm).

6) In the introduction the authors mention that ER-PM junctions serve a physiologically important role. It would be useful for the reader if the authors could elaborate on this somewhat. My understanding is that the existence of these junctions is fascinating, but that not much is yet known about why they are formed or what functional roles they play.

Response: Thank you. This is an important comment. ER-PM junctions were first characterized more than 50 years ago from electron microscopy studies of skeletal and cardiac muscle. These ER-PM junctions in muscle cells provide the basis for excitation-contraction coupling to proceed. Fast forward to 2010's and there has been an explosion of research detailing the importance of ER-PM membrane contact sites for the transfer of lipids, with the identification and characterization of E-Syt proteins, ORP5/8, GRAMD1, TMEM24 etc. It has only been in the last few years that the importance of Kv2.1-mediated ER-PM junctions has become a little clearer. Work from the Trimmer lab has detailed that Kv2.1 containing junctions are essential signaling hubs for Ca²⁺ through their ability to recruit Cav1.2 at the PM and RyR at the ER membrane. We extend this work and provide the first characterization of how these junctions are modified in disease. We have updated the introduction to reflect this knowledge and direct readers to other sources of information to expand their knowledge.

References Cited

1. Wang, Y.-H., Twu, Y.-C., Wang, C.-K., Lin, F.-Z., Lee, C.-Y., and Liao, Y.-J. (2018). Niemann-Pick Type C2 Protein Regulates Free Cholesterol Accumulation and Influences Hepatic Stellate Cell Proliferation and Mitochondrial Respiration Function. *International journal of molecular sciences* *19*, 1678.
2. Misonou, H., Menegola, M., Mohapatra, D.P., Guy, L.K., Park, K.-S., and Trimmer, J.S. (2006). Bidirectional Activity-Dependent Regulation of Neuronal Ion Channel Phosphorylation. *The Journal of Neuroscience* *26*, 13505-13514. 10.1523/jneurosci.3970-06.2006.
3. Nikolic, M., Chou, M.M., Lu, W., Mayer, B.J., and Tsai, L.-H. (1998). The p35/Cdk5 kinase is a neuron-specific Rac effector that inhibits Pak1 activity. *Nature* *395*, 194-198. 10.1038/26034.
4. Nikolic, M., Dudek, H., Kwon, Y.T., Ramos, Y.F., and Tsai, L.H. (1996). The cdk5/p35 kinase is essential for neurite outgrowth during neuronal differentiation. *Genes & development* *10*, 816-825. 10.1101/gad.10.7.816.
5. Dickson, E.J., Jensen, J.B., Vivas, O., Kruse, M., Traynor-Kaplan, A.E., and Hille, B. (2016). Dynamic formation of ER-PM junctions presents a lipid phosphatase to regulate phosphoinositides. *J Cell Biol* *213*, 33-48. 10.1083/jcb.201508106.
6. Castellano, B.M., Thelen, A.M., Moldavski, O., Feltes, M., van der Welle, R.E.N., Mydock-McGrane, L., Jiang, X., van Eijkeren, R.J., Davis, O.B., Louie, S.M., et al. (2017). Lysosomal cholesterol activates mTORC1 via an SLC38A9–Niemann-Pick C1 signaling complex. *Science* *355*, 1306-1311. 10.1126/science.aag1417.
7. Schulien, A.J., Yeh, C.-Y., Orange, B.N., Pav, O.J., Hopkins, M.P., Moutal, A., Khanna, R., Sun, D., Justice, J.A., and Aizenman, E. (2020). Targeted disruption of Kv2.1-VAPA association provides neuroprotection against ischemic stroke in mice by declustering Kv2.1 channels. *Science Advances* *6*, eaaz8110. 10.1126/sciadv.aaz8110.
8. Kutchukian, C., Vivas, O., Casas, M., Jones, J.G., Tiscione, S.A., Simó, S., Ory, D.S., Dixon, R.E., and Dickson, E.J. (2021). NPC1 regulates the distribution of phosphatidylinositol 4-kinases at Golgi and lysosomal membranes. *Embo j*, e105990. 10.15252/embj.2020105990.
9. Tiscione, S.A., Casas, M., Horvath, J.D., Lam, V., Hino, K., Ory, D.S., Santana, L.F., Simó, S., Dixon, R.E., and Dickson, E.J. (2021). IP(3)R-driven increases in mitochondrial Ca(2+) promote neuronal death in NPC disease. *Proc Natl Acad Sci U S A* *118*. 10.1073/pnas.2110629118.
10. Tiscione, S.A., Vivas, O., Ginsburg, K.S., Bers, D.M., Ory, D.S., Santana, L.F., Dixon, R.E., and Dickson, E.J. (2019). Disease-associated mutations in Niemann-Pick type C1 alter ER calcium signaling and neuronal plasticity. *J Cell Biol* *218*, 4141-4156. 10.1083/jcb.201903018.
11. Vivas, O., Tiscione, S.A., Dixon, R.E., Ory, D.S., and Dickson, E.J. (2019). Niemann-Pick Type C Disease Reveals a Link between Lysosomal Cholesterol and PtdIns(4,5)P2 That Regulates Neuronal Excitability. *Cell reports* *27*, 2636-2648.e2634. 10.1016/j.celrep.2019.04.099.

12. Dixon, R.E., Moreno, C.M., Yuan, C., Opitz-Araya, X., Binder, M.D., Navedo, M.F., and Santana, L.F. (2015). Graded Ca^{2+} /calmodulin-dependent coupling of voltage-gated CaV1.2 channels. *Elife* 4. 10.7554/eLife.05608.
13. Vierra, N.C., Kirmiz, M., van der List, D., Santana, L.F., and Trimmer, J.S. (2019). Kv2.1 mediates spatial and functional coupling of L-type calcium channels and ryanodine receptors in mammalian neurons. *Elife* 8. 10.7554/eLife.49953.
14. Vierra, N.C., O'Dwyer, S.C., Matsumoto, C., Santana, L.F., and Trimmer, J.S. (2021). Regulation of neuronal excitation-transcription coupling by Kv2.1-induced clustering of somatic L-type Ca^{2+} channels at ER-PM junctions. *Proceedings of the National Academy of Sciences of the United States of America* 118. ARTN e2110094118
10.1073/pnas.2110094118.
15. Tiscione, S.A.V., O.; Ginsburg, K.S.; Bers, D.M.; Ory, D.S.; Santana, L.F.; Dixon, R.E.; Dickson, E.J. (2019). Disease-associated mutations in Niemann-Pick Type C1 (NPC1) alter ER calcium signaling and neuronal plasticity *Journal of Cell Biology*.

REVIEWERS' COMMENTS

Reviewer #1 (Remarks to the Author):

This is a really interesting, well-designed and thorough study. The findings are entirely novel and timely and will be of wide interest. The authors have addressed all my concerns apart from a minor comment on phrasing:

On p15, line 427 of the revised document, it is (still) stated that the results demonstrate a novel role for NPC1 in organising spatial and functional coupling between PM Kv2.1-Cav1.2 and the ER, which doesn't seem quite right to me - I'd be happy with novel role for NPC1 in downstream organisation of spatial and functional coupling...

Apart from that, congratulations on a very nice study!

Reviewer #2 (Remarks to the Author):

The authors performed significant amount of work and provided additional data for the paper. In my opinion most of the reviewer comments have been satisfactory addressed in the revised manuscript.

Reviewer #3 (Remarks to the Author):

The authors have done a nice job of addressing the concerns and suggestions of the reviewers, and I have no further comments on the revised manuscript.

For future reference, highlighting the changes to the text is a great idea, but then the authors need to do this carefully and highlight all the changes. The new sections on blinding for example are not highlighted.

Comments to the Reviewers

Casas et al.,
NPC1-dependent alterations in Kv2.1–Cav1 nanodomains drive neurodegeneration
NCOMMS-22-33515A

Second revision

We thank all three reviewers for reading our revised manuscript. We greatly appreciate their kind comments about our work. Reviewer comments have improved the rigor of the investigations.

Of the three reviewer's only reviewer 1 had one minor comment remaining which we have addressed in full:

Reviewer 1

Comment: On p15, line 427 of the revised document, it is (still) stated that the results demonstrate a novel role for NPC1 in organising spatial and functional coupling between PM Kv2.1-Cav1.2 and the ER, which doesn't seem quite right to me - I'd be happy with novel role for NPC1 in downstream organisation of spatial and functional coupling...

Reply: thank you. We have taken the reviewer's excellent suggestion and edited the sentence to read: *"Taken together, these results demonstrate a role for NPC1 in downstream organization of spatial and functional coupling between PM Kv2.1–Cav1.2 and the ER to regulate ER–PM Ca²⁺ nanodomains"*.